# Joint COVID-19 and influenza-like illness forecasts in the United States using internet search information

Simin Ma [1], Shaoyang Ning[2] & Shihao Yang [1✉]

## Abstract

**Background** As the prolonged COVID-19 pandemic continues, severe seasonal Influenza (flu) may happen alongside COVID-19. This could cause a "twindemic", in which there are additional burdens on health care resources and public safety compared to those occurring in the presence of a single infection. Amidst the raising trend of co-infections of the two diseases, forecasting both Influenza-like Illness (ILI) outbreaks and COVID-19 waves in a reliable and timely manner becomes more urgent than ever. Accurate and real-time joint prediction of the twindemic aids public health organizations and policymakers in adequate preparation and decision making. However, in the current pandemic, existing ILI and COVID-19 forecasting models face shortcomings under complex inter-disease dynamics, particularly due to the similarities in symptoms and healthcare-seeking patterns of the two diseases.

**Methods** Inspired by the interconnection between ILI and COVID-19 activities, we combine related internet search and bi-disease time series information for the U.S. national level and state level forecasts. Our proposed ARGOX-Joint-Ensemble adopts a new ensemble framework that integrates ILI and COVID-19 disease forecasting models to pool the information between the two diseases and provide joint multi-resolution and multi-target predictions. Through a winner-takes-all ensemble fashion, our framework is able to adaptively select the most predictive COVID-19 or ILI signals.

**Results** In the retrospective evaluation, our model steadily outperforms alternative benchmark methods, and remains competitive with other publicly available models in both point estimates and probabilistic predictions (including intervals).

**Conclusions** The success of our approach illustrates that pooling information between the ILI and COVID-19 leads to improved forecasting models than individual models for either of the disease.

## Plain language summary

Data from the internet enables the presence of infectious diseases such as influenza (flu) to be tracked and monitored. During the ongoing COVID-19 pandemic people will also be infected with flu, impacting health care providers. Predicting both COVID-19 and flu outbreaks in a timely manner enables health care providers and policymakers to prepare for the outbreaks. In this work, we develop a model to jointly predict cases of both COVID-19 and influenza-like illness that can be used at national and state levels in the USA. Our approach is more accurate than alternative similar approaches that predict cases of a single disease, showing the value of predicting the incidence of multiple diseases at the same time.

[1] H. Milton Stewart School of Industrial and Systems Engineering, Georgia Institute of Technology, Atlanta, GA, USA. [2] Department of Mathematics and Statistics, Williams College, Williamstown, MA 01267, USA. ✉email: shihao.yang@isye.gatech.edu

The rising numbers of co-infections of COVID-19 and Influenza (flu)[1,2] have raised serious concerns about the potential of a "twindemic" among the general public[3]. This is also evident in the remarkable similarity between epidemic trends of flu and COVID-19 (Fig. 1). The fast-developing COVID-19 pandemic, coupled with a severe flu season, would overwhelm the already heavily-burdened health care systems, causing further inconceivable losses[4]. This calls for an urgent need to establish an accurate and robust bi-disease tracking/forecasting system to provide public health officials with reliable, timely information to make informed decisions to control and prevent the onset of a "twindemic". To this end, we propose ARGOX-Joint-Ensemble, a principled framework that utilizes the connectivity between flu and COVID-19 to integrate previously proposed forecasting models and adapt to a new era where flu and COVID-19 co-evolve.

Accurate tracking of flu outbreaks and trends is important but non-trivial. In fact, flu affects 9-41 million people annually between 2010-2020 seasons in the United States, resulting in between 12 and 52 thousands of deaths[5]. For decades, the U.S. Centers for Disease Control and Prevention (CDC) monitors flu activities through Influenza-like Illness Surveillance Network (ILINet), which collects the number of outpatients with Influenza-like Illness (ILI) from thousands of healthcare providers and publishes the weekly ILI percentages (%ILI, i.e., the percentages of outpatients with ILI) at the national, regional levels (10 Health and Human Services (HHS) regions in the US), and state levels. However, due to the time required for data collection and administrative processing, the ILI reports from CDC lag behind real time by 1–2 weeks, and thus unable to provide most accurate and timely information on the disease development. Numerous ILI tracking approaches have therefore been proposed, utilizing statistical models[6,7], mechanistic models such as compartmental models[8–11], ensemble approaches[12], and deep learning models[13,14]. Several approaches rely on external signals such as environmental conditions and weather reports[15,16]; social media, such as Twitter posts[17,18] and Wikipedia article views[19,20]; search engine data, such as: Google[21–25], Yahoo[26], and Baidu internet searches[27].

Similarly, many ILI forecasting approaches are adapted and modified to predict the newly emerged COVID-19 pandemic[8,28]. In particular, machine learning (data-driven) methods[28–30] and compartmental models[31–33] are the most popular and prevailing approaches for the publicly-available COVID-19 spread forecasts, according to the weekly forecast reports compiled by CDC[34]. Yet, they also do not capture the inter-correlation between the two

diseases, which could be a crucial factor as both infectious diseases co-evolve.

Evidently, COVID-19 is very likely to circulate for a long period of time and co-evolve with ILI, especially when COVID-19 variants continue to evolve[1]. Hence, a unified robust forecasting framework for both diseases is eminently indispensable.

Despite the development in the methodology tracking individual diseases, joint tracking of flu and COVID-19 remains challenging. In the midst of the on-going COVID-19 pandemic, %ILI collected by CDC may get "contaminated" in the current season, due to symptomatic similarities with COVID-19 as well as various biological and demographic factors. On the other hand, ILI outbreaks can potentially assist COVID-19 cases and deaths predictions, due to the proximity between the two diseases. However, the inter-correlation between COVID-19 and ILI is latent and varies across geographical areas, which can be challenging to capture and utilize for forecasts.

Few attempts have been made to study the connection between COVID-19 and ILI trends, or to incorporate their simultaneous growths for forecasting, while considering the geographical dependence structure (at the state-level). Most of the existing works adapted ILI forecasting model framework and applied towards COVID-19 predictions, or vise-versa. For example, ref. [35] studies the ILI vaccination rates' correlation with COVID-19 deaths, and states its potential prediction power of deaths' trends. ref. [36] extends this study to identify association between vaccination rates and COVID-19 infection, deaths and hospitalization, as well as arguing for their forecasting potentials. ref. [37] uses incidence patterns from past flu seasons, COVID-19 time series information, and demographic covariates in a Generalized Linear Model to forecast next week's county-level case counts, under mild assumptions on the similarity of the transmission mechanisms between COVID-19 and flu. ref. [38], on the other hand, explores seasonal similarities between historical flu seasons and current COVID-19 related signals using a deep clustering module (learn lower-dimensional representation of the signals and reconstruct for forecasting using attention), and produces 1 week ahead independent state-level ILI forecasts.

Inspired by the affinity between ILI and COVID-19's growth trends (Fig. 1), we propose to leverage external COVID-related signals (confirmed cases), along with relevant public search information, for Influenza-like Illness (%ILI) forecasts, and vise-versa for COVID-19 cases and deaths predictions. Yet, to build a COVID-ILI joint prediction model with online search data, many challenges remain to be addressed. For example, the COVID-ILI co-evolution is a new phenomenon, with limited external signals,

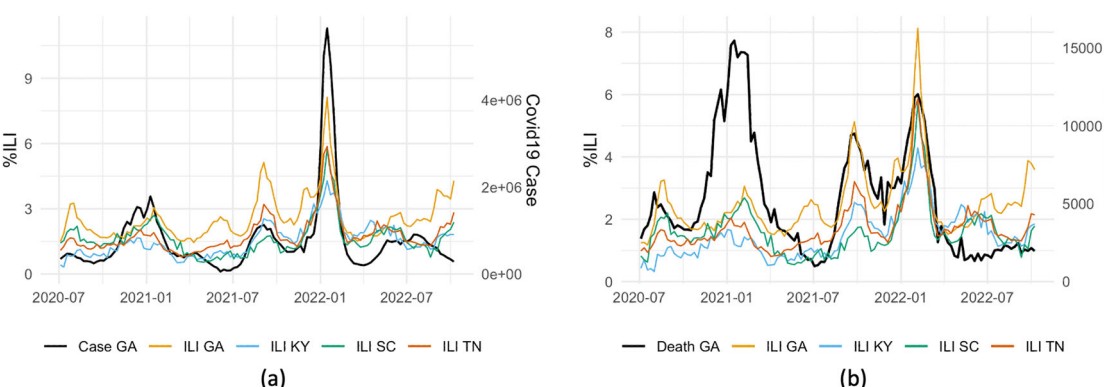

**Fig. 1 Illustration of Georgia (GA)'s real-time COVID-19 cases/deaths (black) growth in comparision with its own lagged %ILI (yellow) and the lagged %ILI of the neighboring states, from 2020-07-04 to 2022-08-13. a** COVID-19 real-time cases in GA (thick black curve) vs. lagged 1 week %ILI in GA (yellow curve) and the neighboring states; (**b**) COVID-19 real-time deaths in GA (thick black curve) and lagged 3 weeks %ILI in GA (yellow curve) and the neighboring states. The underlying data is found in Supplementary Data 1.

while relevant internet search information can be noisy and unstable; hence, it would be a great challenge to efficiently learn the model under data paucity and data instability.

Here we propose ARGOX-Joint-Ensemble, a principled way to integrate and adapt previously proposed flu and COVID-19 forecasting models to "unseen" scenarios where flu and COVID co-exist. In particular, we modified previously proposed forecasting models by incorporating COVID-19 signals for flu predictions and vise-versa for COVID-19 forecasts. We consolidated the models for two diseases through a spatial-temporal fashion to efficiently capture and incorporate COVID-ILI signals for state-level forecasts, while maintaining model features for national-level forecasts. Finally, we employ an ensemble approach to efficiently combine COVID and flu forecasting methods into one joint framework, which is able to effectively shift focuses between COVID and ILI signals for both diseases' forecasts, and produce robust forecasts despite unstable search information signals as inputs. The ensemble framework is systematic and comprehensive. Each data-driven sub-model within the framework is intentionally straightforward and unified to prevent over-fitting. Numerical comparisons show that our method performs competitively with other publicly available single-disease forecasting methods. This study further emphasizes the general applicability and the predictive power of online search data for various tasks in disease surveillance.

## Methods

**Data acquisition and pre-processing**. This paper focuses on the 50 states of the United States, plus Washington D.C for COVID-19 cases and deaths forecasting, while excluding Florida (whose ILI data is not available from CDC) and including New York City and Washington D.C. for %ILI forecasting. For COVID-19 cases and deaths forecasting, we use confirmed cases, confirmed deaths, confirmed new hospital admissions (hospitalization), ILI and Google search query frequencies as inputs. For %ILI forecasting, we use lagged %ILI, COVID-19 cases, and Google search query frequencies as inputs.

*COVID-19 reporting data*. We use reported COVID-19 confirmed cases and deaths of United States from New York Times (NYT)[39] as features in our model. We also use COVID-19 confirmed new hospital admissions (hospitalization) released by U.S. Department of Health and Human Services (HHS)[40] as features for our COVID-19 death forecasts. When comparing against other benchmark methods published in CDC COVID-19 Forecast Hub[34], we use COVID-19 confirmed cases and deaths from JHU CSSE COVID-19 dataset[41], a curated dataset used by the CDC at their official website, as the groundtruth. We do not use JHU COVID-19 dataset as input features in our model because JHU COVID-19 dataset retrospectively corrects past confirmed cases and deaths due to reporting error or changes in federal and state policies. NYT dataset, on the other hand, does not revise past data, which gives more realistic forecasts based on the real-time. All data sources are collected from January 21, 2020 to August 13, 2022.

*CDC's ILINet data*. CDC releases a report of %ILI for the previous week every Friday, which contains the percent of outpatient visits with influenza-like illness for the whole nation, 10 HHS regions, 50 states (except Florida), Washington DC, and New York City (separated from New York State)[42]. CDC's %ILI data for this study are collected from January 21, 2020 to August 13, 2022.

*Google search data*. The online search data used in this paper is obtained from Google Trends[43], where one can obtain the search frequencies of a term of interest in a specific region, time frame, and time frequency by typing in the search query on the website. With Google Trends API, we are able to obtain a daily time series of the search frequencies for the term of interest, including all searches that contain all of its words (un-normalized)[43].

We use 23 highly correlated COVID-19 related Google search queries discovered in prior study[44] (in daily frequency) for COVID-19 cases and deaths forecasts, while using ILI related queries (weekly frequency) from previous study[22,24] for %ILI forecasts. We obtain the search queries for national, regional (summation from states) and state level. For COVID-19 forecasts, we follow the prior work's data cleaning procedures[44], and find the optimal lag of each Google search query from COVID-19 cases/deaths[44] (shown in Table S3 in Supplementary Tables) as inputs to the forecasting models. Figure S1a and S2b (Supplementary Figures) show that the peak of COVID-19 search volume for query "loss of taste" ahead of the peak in reported cases and deaths, confirming strong connections between people's search behaviors and COVID-19 trends.

*%ILI data imputation*. %ILI is weekly indexed while COVID-19 cases and deaths are daily indexed. As we propose a joint forecast framework for both COVID-19 cases/deaths and %ILI in this study, the discrepancy in time stamps between the two needs to be resolved. For this study, we impute daily %ILI as the same number as weekly %ILI, assuming the daily proportion of patients with ILI symptoms is consistent with the weekly number. Imputing daily data also enables larger training sets. We also included a sensitivity analysis in Table S10 (Supplementary Tables).

### Forecasting methods

*National level*. We propose a joint framework for national level COVID-19 cases and deaths prediction, by additionally incorporating flu information in the previously proposed national COVID-19 forecast model[44]. Similarly, we also include COVID-19 cases information for %ILI predictions in the Influenza-like Illness forecast model[22]. Both of the COVID-19 and ILI models are based on the ARGO (AutoRegressive with exogenous GOogle search) method.

Specifically, motivated by the robust performance of ARGO method[44] and the connection between COVID-19 cases/deaths and lagged %ILI (Fig. 1), we add lagged daily imputed %ILI information in the $L_1$ penalized LASSO regression as extra exogenous variables to produce future 28 days' COVID-19 cases and death predictions. That is, we use lagged cases, Google search and ILI information as exogenous variables for COVID-19 cases forecasts, and use lagged hospitalization, deaths, Google search and ILI information for COVID-19 death forecasts. Then, we aggregate the daily predictions into future 4 weeks ahead forecasts for reporting and evaluation, consistent with other publicly available benchmark methods. Meanwhile for ILI, we obtain accurate estimates of 1–2 weeks ahead national %ILI using the ARGO method[22], by additionally incorporating national COVID-19 cases (weekly aggregated) as exogenous variables. Detailed regression formulations are included in the Supplementary Methods section "ARGO-Nat Prediction". We denote this method as bi-disease "ARGO-Nat" method, where "Nat" means national-level.

*State level*. To handle the complicated disease dynamic when COVID-ILI co-evolves, we propose a new ensemble framework, "ARGOX-Joint-Ensemble", which uses joint COVID-ILI information to guide previously proposed disease forecasting methods for unified COVID-19 and %ILI state-level forecasting.

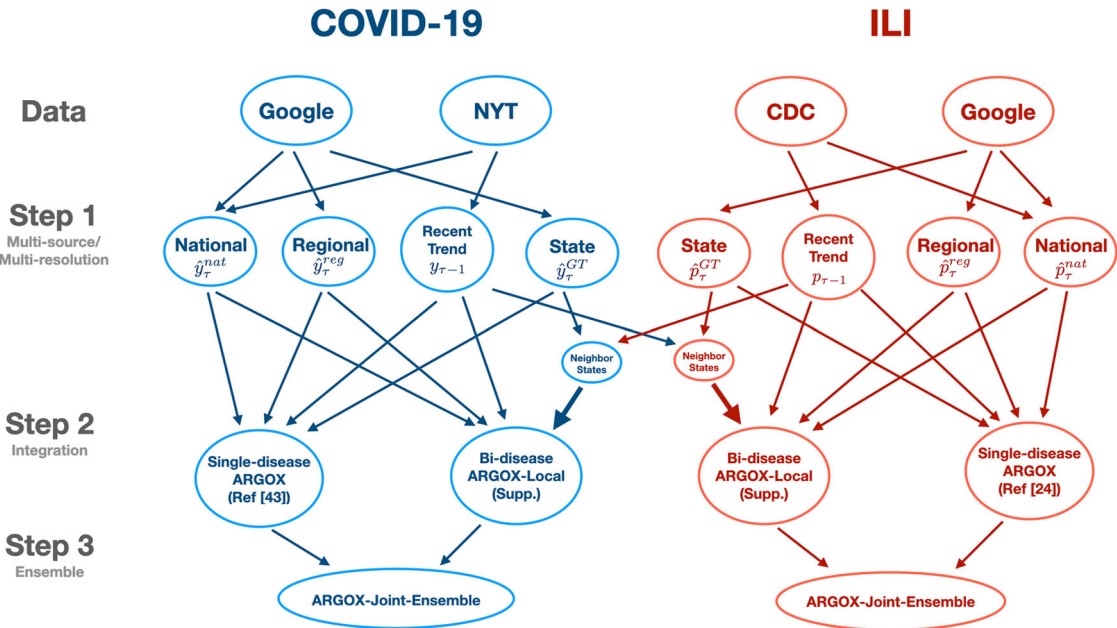

**Fig. 2 Flow Chart of the proposed ARGOX-Joint-Ensemble.** The top-to-bottom procedure of ARGOX-Joint-Ensemble is presented, starting with raw data input and ending with state-level forecasting outputs. The procedures to forecast COVID-19 cases/deaths are in color blue (left), and the procedures to forecast %ILI is in color red (right). Google: Google search data; NYT: New York Times published COVID-19 data.

A high-level illustration of our propose method is shown in Fig. 2, where ARGOX-Joint-Ensemble operates in 3 steps.

In the first step, we gather the raw estimates of COVID-19 cases/deaths (left of Fig. 2) and raw estimates of %ILI (right of Fig. 2) in different geographical resolution. For COVID-19, our raw estimates for state $m$ week $\tau$ cases/deaths $y_{\tau,m}$ are $\hat{y}_{\tau,m}^{GT}$, $\hat{y}_{\tau,r_m}^{reg}$, $\hat{y}_\tau^{nat}$, and $y_{\tau-1,m}$, where $r_m$ is the region number for state $m$. Here, we denote $GT$ and $reg$ to be state/regional estimates with internet search information only, and $nat$ to be national estimates (same as prior study[44]). Similarly, we obtain the raw estimates for state $m$ weekly %ILI $p_{\tau,m}$: $\hat{p}_{\tau,m}^{GT}$ [24], $\hat{p}_{\tau,r_m}^{reg}$ [23], $\hat{p}_\tau^{nat}$ [22], $p_{\tau-1,m}$.

In the second step, we fit two models separately using the raw estimates from step 1 as inputs. Motivated by the connection between lagged neighboring states' %ILI and real-time COVID-19 trends (Fig. 1), we first propose the bi-disease "ARGOX-Local" method. For COVID-19 cases/deaths predictions, bi-disease ARGOX-Local incorporates neighboring state's %ILI information; similarly for %ILI predictions, bi-disease ARGOX-Local includes neighboring state's COVID-19 cases. Besides bi-disease ARGOX-Local, we also directly employ the previously proposed single-disease forecasting models for COVID-19[44] and %ILI[24] in the second step, since they have already demonstrated robust results prior to the newly emerged bi-disease dynamics.

In the third (last) step, we gather the two methods in step 2, to produce the final winner-takes-all ensemble predictions for future 4 weeks COVID-19 cases/deaths and future 2 weeks %ILI. Particularly, for a training period of (overlapping) 15 weeks, we evaluate both predictors (from two models in the second step) with mean squared error (MSE) and select the one with lowest MSE as the ensemble predictor for future weeks.

Implementation details about bi-disease ARGOX-Local, and the final ensemble step ARGOX-Joint-Ensemble, as well as the modifications on previously proposed single-disease forecasting models for COVID-19[44] and %ILI[24], are presented in the Supplementary Methods section "Newly Proposed Bi-disease ARGOX-Local". Detailed ARGOX-Joint-Ensemble's prediction interval calculation is also included in the Supplementary Methods section "ARGOX-Joint-Ensemble".

**Reporting summary**. Further information on research design is available in the Nature Portfolio Reporting Summary linked to this article.

## Results

In this section, we conduct retrospective estimation of the 1–4 weeks ahead COVID-19 cases and deaths, and 1-2 weeks ahead %ILI, at the US national and state level for the period of July 4, 2020 to August 13, 2022. We analyze our joint framework's performances by conducting comparison analysis with our own methods, as well as with other publicly available methods from CDC Forecast Hub[34,45–54] for COVID-19 forecasts. CDC official predictions are a compilation of predictions from all teams that submit their weekly predictions every Monday since January 15th 2020, contributed by different research groups or individuals. Among over 100 teams submitted to CDC, we consider top 6 CDC-published teams for our COVID-19 cases and deaths prediction comparisons. The teams considered in the COVID-19 cases' and deaths' retrospective comparisons are slightly different, due to the difference in teams on the forecasting targets, predictions availability, and studied states/periods. We set the COVID-19 cases and deaths forecasting horizon to be 4 weeks, following CDC Forecast Hub's guidelines[34] and the ease of conducting retrospective comparison. On the other hand, we set %ILI forecasting horizon to be 2 weeks, following previously proposed methods[22–24] and other data-driven %ILI forecasting studies[12,25,28,38].

For COVID-19 cases and deaths forecast, we focus on United State's 51 states/districts (including Washington DC). For %ILI forecast, we focus on United State's 51 states/districts/cities (including Washing DC and New York City, and excluding Florida).

We use three metrics to evaluate the accuracy of a point estimate of COVID-19 cases/deaths or %ILI against the actual COVID-19 cases/death (published by JHU) or %ILI (published by CDC) respectively: the root mean squared error (RMSE), the mean absolute error (MAE), and the Pearson correlation (Correlation). RMSE between an estimate $\hat{y}_t$ and the true value $y_t$ over

**Table 1 Comparison of different methods for state-level COVID-19 1–4 weeks ahead deaths predictions in 51 U.S. states.**

|  | 1 week ahead | 2 weeks ahead | 3 weeks ahead | 4 weeks ahead |
|---|---|---|---|---|
| **RMSE** | | | | |
| ARGOX-Joint-Ensemble | **79.05** | **96.93** | **127.95** | **146.75** |
| ● Ref. [44] | 91.71 | 99.26 | 131.25 | 153.98 |
| ● ARGOX-Local | 84.53 | 103.77 | 133.89 | 161.33 |
| Naive | 89.47 | 102.91 | 129.09 | 148.35 |
| **MAE** | | | | |
| ARGOX-Joint-Ensemble | **44.51** | **56.97** | **66.89** | **79.95** |
| ● Ref. [44] | 46.76 | 60.44 | 69.92 | 89.76 |
| ● ARGOX-Local | 48.44 | 63.45 | 89.10 | 109.41 |
| Naive | 48.64 | 62.75 | 78.61 | 93.86 |
| **Correlation** | | | | |
| ARGOX-Joint-Ensemble | **0.82** | **0.79** | **0.77** | **0.72** |
| ● Ref. [44] | 0.79 | 0.76 | 0.72 | 0.69 |
| ● ARGOX-Local | 0.73 | 0.56 | 0.44 | 0.34 |
| Naive | 0.74 | 0.68 | 0.59 | 0.50 |

The averaged RMSE, MAE, and correlation across 51 states are reported for each forecasting horizon and best performed method is highlighted in boldface. On average (across forecasting horizon), ARGOX-Joint-Ensemble achieves around 6.3% RMSE, 7.2% MAE, and 4% Correlation improvements from ref. [44] for 1–4 weeks ahead prediction.

period $t = 1, \ldots, T$ is $\sqrt{\frac{1}{T}\sum_{t=1}^{T}(\hat{y}_t - y_t)^2}$. MAE between an estimate $\hat{y}_t$ and the true value $y_t$ over period $t = 1, \ldots, T$ is $\frac{1}{T}\sum_{t=1}^{T}|\hat{y}_t - y_t|$. Correlation is the Pearson correlation coefficient between $\hat{\boldsymbol{y}} = (\hat{y}_1, \ldots, \hat{y}_T)$ and $\boldsymbol{y} = (y_1, \ldots, y_T)$.

We use two metrics to evaluate the accuracy of probabilistic predictions and prediction intervals of COVID-19 cases/deaths or %ILI against the actual number published by JHU (for COVID-19) or CDC(for %ILI): the weighted interval score (WIS) and the empirical coverage. The weighted interval score (WIS)[55] is a proper scoring rule (smaller is better), that takes an entire predictive distribution into account and penalises over- and under-confidence. Following CDC Forecast Hub's submission guideline[34], in this study, the WIS between the true value $y_t$ and a predictive distribution at time $t$ is evaluated across 11 prediction intervals with $\alpha_1 = 0.02$, $\alpha_2 = 0.05$, $\alpha_3 = 0.1$, ..., $\alpha_{11} = 0.9$ (implying nominal coverages of 98%, 95%, 90%, …, 10%). The WIS between the true value $\boldsymbol{y} = (y_1, \ldots, y_T)$ and predictive distributions over period $t = 1, \ldots, T$ is computed by averaging the WIS across the period $t = 1, \ldots, T$. The empirical coverage between the true value $\boldsymbol{y} = (y_1, \ldots, y_T)$ and the prediction intervals over period $t = 1, \ldots, T$ is the proportion of true values falling inside a given central prediction interval with 95% nominal coverage.

**COVID-19 deaths**. For national level analysis, we compare ARGO-Nat with two other baseline models (i) persistence (Naive) and (ii) vanilla ARGO prediction[44], where the ground truth is the actual COVID-19 weekly deaths released by JHU dataset[41]. Vanilla ARGO predictions at the national level are obtained from a $L_1$-penalized regression, utilizing the lagged COVID-19 hospitalization, deaths and optimal lagged Google search terms, while the ARGO-Nat prediction further added lagged national (imputed) ILI as exogenous variables. The Naive (persistence) predictions use current week's deaths' counts from New York Times (NYT) as next 1 to 4 weeks' estimation.

Figure S4 and Table S11 (Supplementary Figures and Supplementary Tables) compare the national level estimates against the true COVID-19 weekly deaths. ARGO-Nat estimations improve upon the previous-proposed ARGO method (ref. [44]) in almost all prediction horizons with the help of lagged %ILI information serving as extra exogenous features (see Fig. S2 in Supplementary Figures for the coefficient heatmap). Lagged flu information prevents the previous ARGO framework from

overshooting, and helps with the speedy recovery from estimation spikes, shown in Fig. S4, especially over the three rapid changing and increasing periods (December 2020 to March 2021, July 2021 to November 2021, and December 2021 to March 2022). However, the connection between %ILI information and COVID-19 deaths (target) deteriorates as prediction horizon extends to 3 and 4 weeks ahead (see Table S5). Also, the improvement is limited from ref. [44], as the search terms and time series' signals are saturated and the previous ARGO method is already doing a good job on national level (see Fig. S4). This is also shown in Table S12 (Supplementary Tables), when comparing ARGO-Nat estimations with other publicly available methods released by CDC. Overall, ARGO-Nat estimations is able to achieve competitive performance on national level with subtle improvement over ARGO-Inspired (ref. [44]).

For state-level sensitivity analysis, we compare ARGOX-Joint-Ensemble with three other methods: (i) persistence (Naive) predictions, (ii) single-disease ARGOX method (ref. [44] state-level predictions), (iii) bi-disease ARGOX-Local method (Methods section). Specifically, ref. [44] predictions for each state are obtained from a ARGOX-based ensemble without ILI information, whereas ARGOX-Joint-Ensemble adds the additional bi-disease ARGOX-Local method (Methods section) to produce an ensemble that incorporates ILI information.

Table 1 summarizes the overall results of the comparing methods, averaging over the 51 states for the whole period of July 4, 2020 to August 13, 2022. Our ARGOX-Joint-Ensemble method improves upon ref. [44] thanks to bi-disease ARGOX-Local, which is a strong predictor by gathering neighboring states' deaths and %ILI information as additional feature in the spatial-temporal structure similar to the vector auto-regressive model with exogenous variable (VAR-X). Unlike the other predictors in the ensemble framework, bi-disease ARGOX-Local focuses on each individual state locally while utilizing its neighboring states' information, which shows its power in the ensemble framework as the pandemic progresses. This is also shown in the detailed break-down of methods contributing to ARGOX-Joint-Ensemble in Table S4 (Supplementary Tables). ARGOX-Local is being selected the most on average (around 35%) in the winner-takes-all ensemble. Other predictors also contribute heavily to the ensemble approach and together they provide us a unified COVID-ILI tracking framework with improved robustness and accuracy. Table 2 further compares against other CDC published teams, and the ARGOX-Joint-Ensemble is among the top three

**Table 2 Comparison among different models' 1 to 4 weeks ahead U.S. states level weekly deaths predictions (from 2020-07-04 to 2022-08-13).**

|  | 1 week ahead | 2 weeks ahead | 3 weeks ahead | 4 weeks ahead | Average |
|---|---|---|---|---|---|
| **RMSE** | | | | | |
| COVIDhub-ensemble[34] | 73.42 | 82.54 | 93.32 | 106.45 | 88.93 |
| ARGOX-Joint-Ensemble | (#3)79.05 | (#3)97.93 | (#3)127.95 | (#2)146.75 | (#2)112.92 |
| UMass-MechBayes[45] | 78.48 | 96.11 | 122.04 | 164.17 | 115.20 |
| Naive | 89.47 | 102.91 | 129.09 | 148.35 | 117.45 |
| MOBS-GLEAM COVID[46] | 93.94 | 109.30 | 131.05 | 148.90 | 120.80 |
| LANL-GrowthRate[47] | 191.02 | 104.57 | 119.36 | 134.84 | 137.45 |
| UA-EpiCovDA[48] | 195.70 | 106.83 | 120.44 | 128.88 | 137.96 |
| epiforecasts-ensemble[49] | 217.73 | 151.61 | 155.24 | 158.79 | 170.85 |
| **MAE** | | | | | |
| COVIDhub-ensemble[34] | 39.88 | 46.61 | 54.48 | 63.55 | 51.13 |
| UMass-MechBayes[45] | 42.69 | 53.04 | 65.24 | 82.99 | 60.99 |
| ARGOX-Joint-Ensemble | (#3)44.51 | (#3)56.97 | (#3)66.89 | (#2)79.95 | (#3)62.08 |
| LANL-GrowthRate[47] | 59.97 | 60.55 | 71.20 | 83.49 | 68.80 |
| UA-EpiCovDA[48] | 61.47 | 63.47 | 72.64 | 80.84 | 69.61 |
| Naive | 48.64 | 62.75 | 78.61 | 93.86 | 70.97 |
| MOBS-GLEAM COVID[46] | 49.76 | 63.79 | 79.03 | 93.92 | 71.63 |
| epiforecasts-ensemble1[49] | 64.11 | 66.75 | 77.10 | 88.49 | 74.11 |
| **Correlation** | | | | | |
| ARGOX-Joint-Ensemble | (#1)0.82 | (#1)0.79 | (#1)0.77 | (#1)0.72 | (#1)0.77 |
| COVIDhub-ensemble[34] | 0.80 | 0.77 | 0.72 | 0.68 | 0.74 |
| UMass-MechBayes[45] | 0.78 | 0.72 | 0.66 | 0.60 | 0.69 |
| LANL-GrowthRate[47] | 0.74 | 0.69 | 0.64 | 0.58 | 0.66 |
| epiforecasts-ensemble1[49] | 0.72 | 0.67 | 0.61 | 0.56 | 0.64 |
| UA-EpiCovDA[48] | 0.71 | 0.65 | 0.59 | 0.56 | 0.63 |
| Naive | 0.74 | 0.68 | 0.59 | 0.50 | 0.63 |
| MOBS-GLEAM COVID[46] | 0.72 | 0.66 | 0.57 | 0.49 | 0.61 |

The RMSE, MAE, Pearson correlation and their averages are reported. Methods are sorted based on their average. Our ARGOX-Joint-Ensemble's ranking for each error metric are included in parenthesis. We only show top 6 benchmark models among the 100+ models submitted to the CDC

models, throughout the evaluation period. We also examine the model performance during different periods with rapidly changing dynamics when forecasts are most challenging. Table S5 (Supplementary Tables) shows the forecasting performances for three specific periods: COVID-19 second wave (November 2020 to March 2021), COVID-19 Delta variant (July 2021 to November 2021), and COVID-19 Omicron variant (January 2022 to March 2022). ARGOX-Joint-Ensemble is still able to produce accurate short-term forecasts, and reasonable 3–4 weeks forecasts.

To further examine the ARGOX-Joint-Ensemble's performance gain, we zoom in on the forecasting performances of the two U.S. states, Georgia (GA) and North Carolina (NC) (Table S16 and S18 in Supplementary Tables, and Figs. S12 and S14 in Supplementary Figures). Notably, the bi-disease ARGOX-Local model for COVID-19 is more accurate during the increasing periods (e.g. COVID-19 death from Jul 2021 to Oct 2021) and peaking periods (e.g. COVID-19 death in early Oct 2021 and early Feb 2022) than the single disease model. On the other hand, the bi-disease ARGOX-Local model could overshoot and have delayed recovery after peaking periods (e.g. late Feb 2022 post Omicron peak), possibly due to the misleading %ILI signal. Luckily, the ensemble framework is able to select the more robust one between the single-disease and the bi-disease sub-models. Specifically in GA, the ARGOX-Joint-Ensemble selects the bi-disease ARGOX-Local during the increasing periods (Jul 2021 to Oct 2021, and Jan 2022 to Mar 2022), while "falling back" to single disease sub-model post-peak periods (Oct 2021 to Dec 2021, and Mar 2022 to May 2022). We observe similar patterns in bi-disease model performances and ensemble selection behaviors in NC (Table S18 and Fig. S14 in Supplementary Tables and Supplementary Figures) as well as other states.

In addition to the point estimates, ARGOX-Joint-Ensemble also gives prediction intervals, as recommended by[56]. ARGOX-Joint-Ensemble's prediction interval (PI) is constructed based on the selected method from the ensemble (see Supplementary Methods section "ARGOX-Joint-Ensemble"). Table S7 (Supplementary Tables) shows the prediction intervals' empirical coverage and weighted interval score (WIS)[55], as well as comparisons to other CDC published teams. Table S17 and S19 (Supplementary Tables) further show the prediction intervals' empirical coverage and WIS comparisons to COVIDhub-ensemble[34], zooming into Georgia and North Carolina. Figures S13 and S15 (Supplementary Figures) visualize ARGOX-Joint-Ensemble's death prediction intervals in the two states, respectively. In summary, our nominal 95% prediction interval has an actual 91% coverage and 89% coverage for 1 and 2 weeks ahead predictions (across all states in Table S7 in Supplementary Tables), suggesting reasonable uncertainty quantification albeit slight overconfidence. Our Gaussian-approximated probabilistic forecast is also competitive for 1–2 weeks ahead predictions in WIS compared to other publicly available methods. Zooming into Georgia and North Carolina, we demonstrate the robustness of ARGOX-Joint-Ensemble's interval predictions in face of rapidly changing disease dynamics. However, the performance deteriorates as the prediction horizon extends to 3-4 weeks, as we have noted earlier.

**COVID-19 cases**. For COVID-19 cases analysis, we compare with the same baseline models as those in the COVID-19 deaths analysis above. Figure S7 and Table S13 (Supplementary Tables and Supplementary Figures) compare the national level estimates against the true COVID-19 weekly cases. Ref. [44] and ARGO-Nat both steadily outperform the naive method for 1 and 2 weeks ahead. This demonstrates that, similar to deaths forecasting,

**Table 3 Comparison of different methods for state-level COVID-19 1–4 weeks ahead cases predictions in 51 U.S. states.**

|  | 1 week ahead | 2 weeks ahead | 3 weeks ahead | 4 weeks ahead |
|---|---|---|---|---|
| **RMSE** |  |  |  |  |
| ARGOX-Joint-Ensemble | **7842.82** | **13573.22** | **20390.53** | 24634.36 |
| • Ref. [44] | 7991.17 | 14350.51 | 20791.91 | 25195.74 |
| • ARGOX-Local | 8098.05 | 15097.56 | 21396.01 | 29606.53 |
| Naive | 9356.75 | 15619.13 | 20461.86 | **23727.04** |
| **MAE** |  |  |  |  |
| ARGOX-Joint-Ensemble | **3641.34** | **6246.44** | **8332.36** | **11201.95** |
| • Ref. [44] | 3924.85 | 6818.40 | 9324.75 | 11831.68 |
| • ARGOX-Local | 4008.41 | 7041.23 | 9683.69 | 12328.83 |
| Naive | 4311.78 | 7187.70 | 9740.60 | 11839.87 |
| **Correlation** |  |  |  |  |
| ARGOX-Joint-Ensemble | **0.94** | **0.85** | **0.89** | **0.84** |
| • Ref. [44] | 0.91 | 0.84 | 0.89 | 0.80 |
| • ARGOX-Local | 0.87 | 0.58 | 0.47 | 0.39 |
| Naive | 0.87 | 0.68 | 0.46 | 0.29 |

The averaged RMSE, MAE, and correlation across 51 states are reported for each forecasting horizon and best performed method is highlighted in boldface. On average (across forecasting horizon), ARGOX-Joint-Ensemble achieves around 3% RMSE, 8% MAE, and 2% Correlation improvement from ref. [44].

optimally lagged Google search information and epidemic time series information have strong predictive power on cases trends as well. However, different from COVID deaths, the optimal lags in Google search queries are much shorter from COVID-19 cases (Table S3 in Supplementary Tables). When the forecast horizon extends beyond 2 weeks ahead, the majority of the optimal lags from cases are smaller than the forecast horizon, resulting in Google search terms' signal deterioration, which in turn gives worse prediction performances than 1 and 2 weeks'. Yet, by incorporating %ILI information as additional features, ARGO-Nat is able to rely more on the time series information and lagged %ILI when Google search queries' signals deteriorates (see Fig. S7 in Supplementary Figures). Indeed, ref. [44] fall short against the naive estimation in 3 weeks ahead predictions, while ARGO-Nat is still able to produce steady estimations and to quickly recover from overshooting, especially during the Omicron surges from December 2021 to March 2022 and the possible subsequent surges (from July 2022 to September 2022). This is also shown in Table S14 (Supplementary Tables), when comparing ARGO-Nat estimations with other publicly available methods released by CDC. However, ARGO-Nat barely outperforms the naive method for 3 weeks ahead predictions, and falls short for 4 weeks ahead predictions. The short-term signal of search information is an inherent limitation for any long-term predictions (more details in Discussion).

For state-level prediction, Table 3 summarizes the overall results of the baseline methods averaging over the 51 states, while Table 4 summarizes the overall results when comparing with top CDC published teams. By incorporating neighboring states' flu and cases information, our ARGOX-Joint-Ensemble method improves upon ref. [44] and remains competitive among the top performers released by CDC, demonstrating the robustness of COVID-ILI joint framework on cases prediction. This is further illustrated in Table S6 (Supplementary Tables), which shows the state-level forecasting performance in three selected periods with rapidly changing dynamics: COVID-19 second wave (Oct 2020 to Feb 2021), Delta variant (Jul 2021 to Oct 2021), and Omicron variant (Dec 2021 to Mar 2022). ARGOX-Joint-Ensemble maintains its accuracy and robustness throughout the three rapid increasing and decreasing periods, demonstrating its early-warning detection ability, especially for 1–2 weeks ahead forecasts. By taking a closer look at Georgia (GA) and North Carolina (NC), we can see that bi-disease ARGOX-Local model produces robust early-warning estimates before the increasing

and peaking periods around Sep 2021 and Jan 2022, and is less prone to overestimation during the Omicron variant surge, especially for 1 and 2 weeks ahead forecasts (Fig. S16, S17 in Supplementary Figures). Single-disease model (ref. [44]), on the other hand, performs better around the decreasing period in Nov 2021 and Mar 2022 when the ILI signal is noisy and deteriorating. This qualitative result is similar to the COVID-19 death prediction, and ARGOX-Joint-Ensemble is able to combine the best from the two. Moreover, our prediction intervals for 1 and 2 weeks ahead COVID-19 cases achieve reasonable coverage with WIS comparable to other publicly available methods (Table S8 in Supplementary Tables), demonstrating reliability throughout rapid changing dynamics led by different COVID-19 variants. However, signal deterioration from Internet search information still affects the state-level long-term forecast accuracy and prediction interval coverage (see Tables 3 and 4, and S8 in Supplementary Tables). Overall, our joint COVID-ILI framework is competitive with the top CDC released teams in all error metrics, exhibiting strong short-term state-level cases forecasting while maintains its accuracy compared to the naive method in 3 to 4 weeks forecasting.

**%ILI**. To evaluate the accuracy of our %ILI estimations, we compared the estimates with the actual %ILI released by CDC weeks later, and different benchmark methods for national and state level forecasts.

At the national level, we compare ARGO-Nat (Methods section) with three baseline methods: (i) the persistence (Naive) estimates, which simply uses CDC's reported %ILI of the previous week as the estimate for the current week, (ii) estimates by the lag-3 autoregressive model (AR-3 model), (iii) the previously developed ARGO (ref. [22]) without COVID-19 information, which is a $L_1$-penalized regression on past %ILI and flu-related Google search queries. Figure S10 (Supplementary Figures) displays the estimates against actual CDC-reported %ILI, whereas Table S15 (Supplementary Tables) further summarizes all estimations' performances in three error metrics. Ref. [22] outperforms the naive and AR-3 time series estimates in all error metrics for both 1 and 2 weeks ahead forecasts. ARGO-Nat further improves from the single-disease model (ref. [22]) by capturing the COVID-19 cases trends during the same period, with 9% RMSE, 6.2% MAE and 0.7% Pearson Correlation improvements on average across 1 and 2 weeks ahead estimates. Both ref. [22] and ARGO-Nat are able to overcome delaying effect in the %ILI time series predictions by utilizing the responsive

**Table 4 Comparison among different models' 1 to 4 weeks ahead U.S. states level weekly cases predictions (from 2020-07-04 to 2022-08-13).**

| | 1 week ahead | 2 weeks ahead | 3 weeks ahead | 4 weeks ahead | Average |
|---|---|---|---|---|---|
| **RMSE** | | | | | |
| CU-select[52] | 9713.68 | 12117.15 | 16084.91 | 18647.91 | 14140.91 |
| ARGOX-Joint-Ensemble | (#1)7842.82 | (#2)13573.22 | (#4)20390.53 | (#6)24634.36 | (#2)16610.22 |
| COVIDhub-ensemble[34] | 8326.61 | 15276.36 | 20294.56 | 23466.16 | 16840.92 |
| UVA-Ensemble[51] | 12590.87 | 15852.81 | 19017.51 | 21022.70 | 17120.97 |
| Naive | 9356.75 | 15619.13 | 20461.86 | 23727.04 | 17291.20 |
| Karlen-pypm[54] | 10664.30 | 16627.84 | 21741.59 | 25677.79 | 18677.88 |
| CovidAnalytics-DELPHI[53] | 13572.00 | 17768.43 | 21700.17 | 23709.65 | 19187.56 |
| USC-SI_kJalpha[50] | 12389.33 | 28559.93 | 44429.39 | 46928.03 | 33076.67 |
| **MAE** | | | | | |
| CU-select[52] | 4487.76 | 5947.66 | 7973.62 | 9439.14 | 6962.05 |
| ARGOX-Joint-Ensemble | (#1)3641.34 | (#2)6246.44 | (#2)8332.36 | (#3)11201.95 | (#2)7355.52 |
| COVIDhub-ensemble[34] | 3695.18 | 6668.18 | 9380.44 | 11231.05 | 7743.71 |
| UVA-Ensemble[51] | 5168.23 | 7135.38 | 9105.79 | 10674.69 | 8021.02 |
| Naive | 4311.78 | 7187.70 | 9740.60 | 11839.87 | 8269.99 |
| Karlen-pypm[54] | 4553.61 | 7163.27 | 10027.80 | 12417.84 | 8540.63 |
| CovidAnalytics-DELPHI[53] | 7134.51 | 9165.01 | 11144.80 | 12515.78 | 9990.02 |
| USC-SI_kJalpha[50] | 4458.37 | 8161.65 | 12569.92 | 15156.01 | 10086.49 |
| **Correlation** | | | | | |
| ARGOX-Joint-Ensemble | (#1)0.94 | (#1)0.85 | (#1)0.89 | (#1)0.84 | (#1)0.88 |
| USC-SI_kJalpha[50] | 0.87 | 0.81 | 0.72 | 0.61 | 0.75 |
| CU-select[52] | 0.86 | 0.78 | 0.64 | 0.48 | 0.69 |
| Karlen-pypm[54] | 0.87 | 0.68 | 0.49 | 0.36 | 0.60 |
| COVIDhub-ensemble[34] | 0.90 | 0.69 | 0.48 | 0.31 | 0.59 |
| Naive | 0.87 | 0.68 | 0.46 | 0.29 | 0.57 |
| UVA-Ensemble[51] | 0.82 | 0.62 | 0.46 | 0.33 | 0.56 |
| CovidAnalytics-DELPHI[53] | 0.71 | 0.56 | 0.43 | 0.32 | 0.50 |

The RMSE, MAE, Pearson correlation and their averages are reported. Methods are sorted based on their average. Our ARGOX-Joint-Ensemble's ranking for each error metric are included in parenthesis. We only show top 6 benchmark models among the 100+ models submitted to the CDC.

**Table 5 Comparison of different methods for state-level % ILI 1 and 2 weeks ahead forecasts in 51 U.S. states.**

| | 1 week ahead | 2 weeks ahead |
|---|---|---|
| **RMSE** | | |
| ARGOX-Joint-Ensemble | **0.205** | **0.278** |
| • Ref. [24] | 0.260 | 0.291 |
| • ARGO-Local | 0.248 | 0.305 |
| Naive | 0.273 | 0.413 |
| VAR | 0.380 | 0.542 |
| **MAE** | | |
| ARGOX-Joint-Ensemble | **0.165** | **0.230** |
| • Ref. [24] | 0.178 | 0.279 |
| • ARGOX-Local | 0.175 | 0.291 |
| Naive | 0.183 | 0.308 |
| VAR | 0.264 | 0.478 |
| **Correlation** | | |
| ARGOX-Joint-Ensemble | **0.960** | **0.909** |
| • Ref. [24] | 0.823 | 0.879 |
| • ARGOX-Local | 0.846 | 0.883 |
| Naive | 0.812 | 0.801 |
| VAR | 0.704 | 0.490 |

The averaged RMSE, MAE, and correlation across 51 states are reported for each forecasting horizon and best performed method is highlighted in boldface. On average (across forecasting horizon), ARGOX-Joint-Ensemble achieves around 11% RMSE and 12% MAE and 7% Correlation improvements from ref. [22].

search behavior data, and predict almost perfectly from January 2021 to July 2021. The bi-disease ARGO-Nat is able to further prevent undesired over-predictions comparing to the single disease model (ref. [22]), and "foresees" upcoming increasing trends. For instance, ARGO-Nat is more robust towards detecting the upcoming increasing trends around Aug 2020, Jul 2021, and Jan 2022 (Fig. S10 in Supplementary Figures). In particular, when all the comparing methods exhibit delaying behavior in 2 weeks ahead estimates, the bi-disease ARGO-Nat is able to harness both Google search and

COVID-19 cases information to overcome such delay, especially from Jul 2020 to Jan 2021, and from Feb 2022 to Jul 2022.

At the state level, we compare ARGOX-Joint-Ensemble (Methods section) with four baseline models: (i) the persistence (Naive) estimates, (ii) estimates by the lag-1 vector autoregressive model (VAR model), (iii) a previously developed single-disease ARGOX model (ref. [24]) without COVID-19 information, (iv) bi-disease ARGOX-Local (Methods section), which is a sub-component of ARGOX-Joint-Ensemble. Table 5 shows the comparing methods' performance averaging across the 50 states and NYC, where ARGOX-Joint-Ensemble gives the leading performance uniformly in all metrics. While ref. [24], ARGO-Local, and ARGOX-Joint-Ensemble all uniformly outperform the naive and VAR1 predictions on average, ARGOX-Joint-Ensemble is the only method consistently outperforms the naive estimates in all the states in all error metric. For Georgia, ARGOX-Joint-Ensemble uniformly selects bi-disease ARGOX-Local during the rapid changing dynamics period (with sudden increases and decreases) from Nov 2020 to Jul 2021, as ARGOX-Local captures the affinity between COVID-19 growth and ILI development (Fig. S18 in Supplementary Figures). On the other hand, when bi-disease ARGOX-Local is less accurate for %ILI during the initial Omicron outbreak (around Dec 2021), ARGOX-Joint-Ensemble is able to quickly recover and fall back to single-disease model ref. [24]. Figure S11 (Supplementary Figures) further shows all states' RMSE, MAE and Pearson Correlation in the violin charts, with mean and standard deviations, where the joint COVID-ILI ensemble framework reveals its robustness over geographical variability and extracts a strong combination from the other two ARGOX alternatives. Lastly, Table S9 (Supplementary Figures) shows the coverage and WIS of the ARGOX-Joint-Ensemble's prediction intervals across all 50 states and NYC for 1–2 weeks ahead predictions. Our nominal 95% prediction interval has an actual 93% coverage and 89% coverage on average, for 1 and 2 weeks ahead predictions. Meanwhile, ARGOX-Joint-Ensemble can consistently outperform the baseline time series model in WIS, for both 1 and

2 weeks ahead predictions, further demonstrating its robustness and efficient data-driven signal utilization.

## Discussion

In this paper we propose to jointly forecast COVID-19 and Influenza-like Illness in the United States. At the national level, our ARGO-Nat is built upon previously proposed ARGO[44] method and incorporates COVID-19 cases and %ILI as additional exogenous variables for national level %ILI and COVID-19 cases/ deaths predictions. At the state level, ARGOX-Joint-Ensemble is a unified COVID-ILI forecasting framework that efficiently combines multi-source, multi-resolution, bi-disease information, and provides accurate, reliable, real-time COVID-19 cases, deaths and %ILI forecasts, while most previously proposed methods that utilize internet search information[22,24,44] only focus on single disease forecasting. Motivated by the interconnection between % ILI and COVID-19 cases/deaths trends (Fig. 1) within the state and across neighboring states, this study further proposes bi-disease ARGOX-Local which additionally gathers neighboring states' %ILI information for COVID-19 cases/deaths forecast, and utilizes COVID-19 cases for %ILI forecast. Lastly, by aggregating bi-disease ARGOX-Local and the previously proposed single-disease COVID-19 or %ILI forecasting methods in a winner-takes-all ensemble fashion, ARGOX-Joint-Ensemble provides accurate, reliable real-time COVID-ILI joint prediction at the state level. By incorporating cross-regional and temporal correlation of both COVID-19 and Influenza-like Illness activities, ARGOX-Joint-Ensemble outperforms most benchmark methods and achieves reasonable performance against other publicly available models.

At the national level, the strong performance of ARGO-Nat can be attributed to two factors. First, ARGO-Nat is built upon the established COVID-19[44] and Influenza-like Illness tracking[22] frameworks, both of which have already shown their strength by utilizing temporal auto-correlation and dependence between people search behaviors and the target time series information. Second, ARGO-Nat takes a step further by complementing the established single-disease COVID-19 and Influenza-like Illness forecasting frameworks with the pooled information from both diseases. Specifically, by incorporating lagged Influenza-like Illness and COVID-19 information, ARGO-Nat shows performances improvements for COVID-19 cases/deaths and %ILI future predictions, with fast recovery from over-estimations, and smoother forecasts (especially when forecasting 3-4 weeks ahead for COVID-19 cases and deaths).

At the state level, similar two key factors contribute to our ARGOX-Joint-Ensemble model. First, the introduction of the bi-disease constituent model, ARGOX-Local, effectively combines Google search data, Influenza-like Illness activity data, and COVID-19 cases/deaths to produce state-level COVID-19 cases/ deaths estimates or %ILI estimates. This bi-disease ARGOX-Local is an intermediate method that connects Influenza-like Illness and COVID-19 information in geographical proximity, and is a joint COVID-ILI method that uses each other's growth trend to help the other, which can efficiently detect upcoming surges and peaking periods. Second, to further improve accuracy and robustness, we efficiently combine bi-disease ARGOX-Local with the previously proposed single-disease COVID-19 and %ILI single-disease forecasting methods[24,44] to produce winner-takes-all ensemble (ARGOX-Joint-Ensemble) forecast. These methods are adapted directly from the established Influenza-like Illness and COVID-19 predictions with minimal changes, which reduces the chance of over-fitting. The previously proposed winner-takes-all ensemble framework[44] also naturally incorporates bi-disease ARGOX-Local as one additional constituent model, and is able to

robustly select the "best" estimate among the sub-models during rapid changing dynamics. Furthermore, the ARGOX-Joint-Ensemble is able to outperform the constituent models for all states in all 1 to 4 weeks ahead COVID-19 cases/deaths predictions, and 1–2 weeks ahead %ILI predictions, while remaining its accuracy during the Omicron COVID-19 variant in early 2022. When comparing with other state-of-arts models from CDC for COVID-19 forecasts, our national and state-level models are also able to perform reasonably well, further demonstrating the strength of the joint-disease prediction framework.

Like all big-data-based models, our model has its limitations. ARGO-Nat and ARGOX-Joint-Ensemble's accuracy depends on the reliability and stability of its inputs—Google Trends data, historical %ILI data from CDC, and COVID-19 cases/deaths data from NYT. Since the optimal lags between COVID-19 cases/ deaths and Google search queries have short time-span (Table S3 in Supplementary Tables), information in Google search data deteriorates as forecast horizons expand, which could potentially impact the robustness and accuracy of our 3 and 4 weeks ahead COVID-19 predictions. Similarly, the proposed ARGO-Nat (national-level) and ARGOX-Joint-Ensemble (state-level) gradually lose their predictive power towards %ILI, when the forecasting horizon extends to 3 and 4 weeks and thus we focus on 1–2 weeks ahead %ILI predictions in this study. One hypothesis is that the COVID-19 symptomatic and contagious periods last longer than Influenza-like Illness[57], and thus the COVID-19 time series and related Google search information are more predictable for COVID-19 than those for %ILI for longer forecasting horizons. The long-term forecasts' deterioration is indeed a limitation, due to the data-driven nature of our proposed models, impacting the predictions for the onset and the finish of the disease season. Fortunately, by recognizing the correlation between Influenza-like Illness activities and COVID-19 growth trend at both national and state levels, %ILI and COVID-19 cases/deaths are able to rely more on each other when Google search queries' signal degenerates, which enable robust estimations and fast recoveries from overshooting. Although our current model can steadily outperform benchmark methods, models to further capture long-term COVID-19 trends, and boost long-term forecasting performances could be an interesting future direction.

Another limitation is the retrospective nature of this study. Although we are not using any "forward looking" information that wouldn't be available at the time of prediction to reflect "real-time" performances, the input data sources could still be subject to backfill and revisions. This issue is circumvented for COVID-19 as we use the New York Times (NYT) github COVID-19 dataset[39] as the input to our models, which does not revise past data. Yet, CDC reported %ILI[5], could be subject to revisions, due to reporting delays from health-care providers. However, prior Influenza-like Illness forecasting studies have shown that %ILI back-fill and modifications typically would not impact forecasting performances too much[58,59].

In addition, it should be noted that our estimation targets (treated as the groundtruth), JHU COVID-19 dataset (cases, deaths) and CDC's %ILI, can be unreliable. JHU COVID-19 dataset[41] retrospectively corrects past confirmed cases and deaths due to reporting error, especially during the early stage of COVID-19. Furthermore, due to wide availability of rapid antigen tests, the COVID-19 confirmed case count might be an under-estimation of the true volume of infections[60]. On the other hand, %ILI is only a proxy for the actual flu incidence in the population. First of all, %ILI could exhibit high noise at the state-level, as it is calculated from a sample of outpatient visits with influenza-like symptoms and subjects to retrospective revisions[24]. Moreover, the prolonged COVID-19 pandemic and subsequent changes in

public's health care-seeking behaviors further impact %ILI's representation power of laboratory confirmed influenza, as it will capture visits due to any respiratory pathogen that presents with the symptoms of fever plus cough or sore throat, including influenza, SARS-CoV-2 (COVID-19), and Respiratory Syncytial Virus (RSV)[42]. Nevertheless, under the current pandemic situation, accurate predictions of COVID-19 cases, deaths, and%ILI at both national and state levels are still valuable for optimizing resource allocations, and healthcare interventions. For example, the %ILI surveillance data can still reveal the general trend of influenza activity in a particular region and provide invaluable information for optimizing where, when and what influenza viruses are circulating[42]. Studies investigating alternative indicators for COVID-19 and flu incidence in the population can be found in[61,62]. In addition, CDC FluSight is also investigating additional surveillance components to track seasonal influenza activities, including laboratory-confirmed influenza hospital admissions[63]. Therefore, considering alternative influenza activities' indicators as forecasting targets and/or exogenous information in the model could be an important future direction.

In light of recurrent Influenza-like Illness waves and the prolonged COVID-19 pandemic, accurate joint-disease tracking of epidemic activity at different geographical levels has become more important than ever. Our ARGO-Nat and ARGOX-Joint-Ensemble provide high-precision national and state-level surveillance information, which would enable timely decision making and optimal resource reallocation in the face of a potential twindemic. The reliable estimations by our joint COVID-ILI framework give public more insights into both diseases and can serve as valuable resources for public health officials.

## Data availability
The online search data sets that support the findings of this study are all publicly available. The Google search data is obtained from Google Trends[43], and also deposited to Harvard Dataverse (https://doi.org/10.7910/DVN/PGNBAX)[64]. The downloading date of COVID-19 related Google search data is 2022-08-14 and the downloading date of % ILI related Google search data is 2022-09-11. The COVID-19 confirmed cases and deaths data are publicly available from JHU CSSE COVID-19 dataset[41], while hospitalization data are publicly available from HHS[40] and the official CDC Forecast Hub[34]. %ILI data are publicly available from CDC[5].

The numerical data plotted in Fig. 1 is available in Supplementary Data 1.

## Code availability
The code to reproduce the results shown in this study is available under https://zenodo.org/badge/latestdoi/476787213[65].

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

## Acknowledgements

S.Y. is supported by the National Center for Advancing Translational Sciences of the National Institutes of Health under Award number UL1TR002378. The content is solely the responsibility of the authors and does not necessarily represent the official views of the National Institutes of Health.

## Author contributions

S.M., S.N. and S.Y. designed the research; S.M., S.N. and S.Y. performed the research; S.M. analyzed data; S.M., S.N. and S.Y. wrote the paper.

## Competing interests

The authors declare no competing interests.
