## [Peer Review File · Communications Medicine]

Reviewers' comments:

Reviewer #1 (Remarks to the Author):

The authors aim to build a joint influenza and COVID-19 predictor using search trends for the US to exploit commonalities in symptoms and care seeking for both diseases. Although it's a worthwhile effort, their grasp of existing surveillance streams for these diseases is limited, and there are limited insights derived from similarity in search trends. Further, there are no significant methodological advances other than the choice of exogenous variables. Hence I reject the manuscript in its current form, although I appreciate the authors' effort in exploring this important problem.

In Figure 1, they primarily compare COVID-19 deaths data with ILI% from a chosen state and its neighbors. First of all, as the authors already acknowledge, the ILI curve could be heavily contaminated by COVID incidence, and hence any observations of similarity could just be due to that. Further, the time period of high similarity seems to be the Omicron wave, where the symptom profile deviated from earlier strains (upper vs lower respiratory, for example - <https://www.nature.com/articles/d41586-022-00007-8>). Also the peak in Fall 2021 coincides with an RSV surge that was seen across multiple regions (<https://www.cdc.gov/surveillance/nrevss/rsv/natl-trend.html>), which is combined with ILI, while the Deaths are due to Delta variant. Without these additional contexts, such correlation based methods could be misled during subsequent variant driven waves of COVID, or off-season resurgence in influenza activity (like being seen in recent weeks Apr-May in USA).

The authors do not seem to exploit the HHS hospitalization dataset (<https://healthdata.gov/Hospital/COVID-19-Reported-Patient-Impact-and-Hospital-Capa/g62h-syeh>) which is the basis for COVID and Influenza hospitalization forecasting exercises being coordinated by CDC. In addition to being directly attributable to respective diseases (unlike ILI), they also suffer from less backfill and are more timely due to their daily update cadence.

Further it has been known that recent cases and deaths counts from states for COVID-19 have been heavily influenced by reporting issues. Especially for cases, due to wide availability of rapid antigen tests, it does not truly represent the volume of infections being generated on the ground. Death data has been subject to significant backfill during audits, and are hence unreliable for timely forecasting efforts.

The authors statement that "if CDC had published daily ILI data, it would have exhibited similar in-week seasonality to COVID-19 reported cases" belies their understanding of the two surveillance systems. While the ILI is obtained via an Outpatient network of hospitals, and includes a numerator (patients with ILI symptoms) and denominator (total outpatient visits), the latter is obtained from State Departments of Health which have their own mechanism for collecting COVID-19 case data from testing centers and hospitals within the state. Further, daily imputation of a % time series like ILI, would involve imputing numerator and denominator separately, to be consistent, which the authors do not do.

Finally, as the authors are aware of ForecastHub efforts, providing point estimates for

predictions and non-probabilistic error functions (RMSE, MAE, PC) are limited in utility. Especially with the noise in observed data and uncertainty going forward, probabilistic forecasts are more appropriate. I suggest the authors look into <https://journals.plos.org/plosmedicine/article/comments?id=10.1371/journal.pmed.1003793> for future reporting epidemic forecasting tasks. This also makes it difficult to compare against other methods in the Hub, since they are better evaluated using metrics such as weighted interval score (WIS) and coverage.

Reviewer #2 (Remarks to the Author):

In this analysis, Ma et al. develop and evaluate point-estimate forecasting frameworks that pool information between influenza-like illness (ILI) and COVID-19 surveillance data to produce forecasts for COVID-19 reported cases and deaths and percent ILI from the U.S. ILINet system. The authors retrospectively evaluate performance of these approaches at the national and state-level from early 2020 through early 2022 using RMSE, MAE, and correlation accuracy metrics. Specific suggestions follow:

- From a conceptual point, the authors should be clearer that they are not forecasting influenza but forecasting ILI. ILI is impacted by COVID, influenza, and other respiratory pathogens. Therefore, these are joint forecasts for COVID and ILI, not influenza, and the authors have not supported their statement that there's an "affinity between influenza and COVID-19's growth" but that ILI activity and COVID-19 transmission may be related to each other. In the US since the COVID-19 pandemic began, we have witnessed one season with almost no influenza transmission and one season with historically low influenza transmission. This is likely to have resulted because of behavioral interventions adopted to prevent the spread of COVID-19 and possible viral interference between COVID-19 and influenza.
- For these forecasts to be actionable, the authors should provide probabilistic forecast and measure forecast performance using metrics that assess probabilistic accuracy (like the weighted interval score and prediction interval coverage). Forecasts received by the COVID-19 Forecasting Hub, which the authors compare their forecast to, and CDC's FluSight require probabilistic forecasts.
- ILI and COVID case and death data can be subject to backfill (i.e. modification of reported data from days to weeks ago). The authors should only utilize the data available at the time the forecasts would be generated to ensure the accuracy reflects "real-time" performance.
- It's unclear from the paper's methods what forecasting activities could be conducted "in real-time" vs. what could be done retrospectively. For example, would the ILI data used as an input in the COVID case and death forecasting be available at the time forecasts would be generated or does the delay in ILI data publication prevent that? In addition, what data does the "winner takes all" ensemble evaluation need and are those data available in real-time to support that? It's important that the authors are only using data that would be available at the time the forecasts would have to be made, especially since they are comparing themselves against forecasts in Table 2 that were truly made in "real-time".
- It's not clear why the authors are imputing daily ILI data when the ILI and COVID forecasting targets being compared are at the weekly scale and how they validated their approach was accurate since reported COVID-19 cases can be impacted by state reporting choices. The authors should justify this approach and ensure the additional complexity added by it is adding something significant to this study.

- Flu activity increased after the end of the study period in the US while COVID activity decreased. The authors should extend the study period to capture this period to see how these methods perform.
- The authors should look at performance of the specific methods during periods of rapidly changing dynamics (e.g. peaking periods, increasing periods) to see how the methods perform when forecasts are most challenging.

Reviewer #3 (Remarks to the Author):

Summary

In this manuscript, the authors claim that (separate) COVID-19 and seasonal influenza forecasts can be improved by incorporating both COVID-19 and influenza data into the forecasting models.

To support this claim, the authors introduce several models for forecasting COVID-19 deaths, COVID-19 cases, and CDC influenza-like illness (ILI) data, and evaluate their performance at (USA) state and national levels over the period 2020-07-04 to 2021-03-05.

The authors demonstrate that the COVID-19 case forecasts generated by models that incorporate both COVID-19 and influenza data are competitive with the best-performing models in the CDC COVID-19 forecasting ensemble.

Overall impression

The simultaneous circulation of COVID-19 and seasonal influenza is a major concern for healthcare systems around the world.

We should expect some correlation between COVID-19 cases and influenza-like illnesses, since these pathogens share common modes of transmission, and much of this correlation could presumably be explained by human mobility and mixing.

Accordingly, the major claim in this paper is sensible and, as far as I am aware, has not been investigated in the existing literature.

The results clearly support the authors' claim, and it is impressive that the proposed "ARGOX-Joint-Ensemble" model is competitive with the best-performing models in the CDC ensemble.

Major comments

1. The "winner takes all" approach is not described in enough detail to be reproduced.

The authors state: "particularly, for a training period of 15 weeks, we evaluate both predictor with mean squared error (MSE) and select the one with lowest MSE as the ensemble predictor for future weeks."

While this text makes it clear that the approach selects a single model at each forecast date, and separately for each of the 9 forecast targets (1-4 weeks ahead COVID-19 deaths, 1-4 weeks ahead COVID-19 cases, 1 week ahead %ILI), the only supporting information I could find is Table S2, which shows how often each competing model was selected for each of the 9 forecast targets.

At each forecast week, were the previous 15 weeks used as a training period to identify the best model for each forecast target?

If so, how did this account for forecast targets with fewer than 15 data points (e.g., 4 week ahead targets)?

If not, how are the models evaluated?

2. Why did the %ILI forecasts only have a 1-week horizon, as compared to the 4-week horizon used for the COVID-19 forecasts?

I did not notice an explanation for this decision in the provided manuscript (my apologies if the authors have explained their rationale!).

3. I'm keen to understand more about how the forecast predictions changed as a consequence of using both COVID-19 and %ILI data, and how this is reflected in the reported performance improvements.

It's difficult to assess this by eye from the 160 or so figures included in the supplementary materials, but hopefully the authors can address some of the following questions:

- Are the "ARGOX-Joint-Ensemble" forecasts qualitatively different in some way, as a result of using the pooled COVID-19 and %ILI data?

- Are there certain circumstances or narrow windows in time where the "ARGOX-Joint-Ensemble" forecasts substantially out-perform the non-pooled models, or do they instead consistently perform slightly better than the non-pooled models?

- Under what circumstances did the "ARGOX-Joint-Ensemble" consistently select the pooled models, and under what circumstances did it consistently select the non-pooled models?

- **How rapidly** can the "ARGOX-Joint-Ensemble" forecasts recognise when to switch between the pooled and non-pooled models?

If the authors used a 15-week evaluation period to select the best model for each forecast target, did they also consider shorter evaluation periods to allow the ensemble to switch more rapidly between the competing models?

4. In essence, the questions in the previous comment are all aimed at understanding when, and to what degree, we should place confidence in these forecasts.

An even tougher question to answer, but for which I'm curious to hear the authors' thoughts, is whether the improvements obtained from the "ARGOX-Joint-Ensemble" model can offer earlier warning (or some other form of enhanced information) for healthcare providers?

In particular, for infectious diseases, it can be particularly useful to predict when activity will peak and begin to decrease (and, alternatively, when activity will bottom-out and begin to increase).

Have the authors assessed whether the "ARGOX-Joint-Ensemble" model provides better and/or earlier predictions of when there will be a turning point in the data?

Minor comments

1. It would be useful to highlight the study period in Figure 1.

The authors might also consider including similar plots for the national data?

2. It would be interesting to include the "ARGOX Individual" results in Tables 1-4, so that the reader can see a more nuanced comparison of the incremental gains provided by the pooled data ("ARGOX Individual") and the "winner takes all" approach ("ARGOX-Joint-Ensemble"), without resorting to the supplementary material.

3. A table of contents would help the reader to navigate the 172-page supporting information document.

It appears the authors intended to include a table of contents, since the first page contains the statement "This Supplementary Material is organized as following:", but this is not followed by a table or list.

4. Am I correct in understanding that the lags in Table S1 are reported in days?

Reproducibility and availability of materials.

In the Editorial Policy Checklist, the authors have indicated that all relevant code is provided and that the manuscript includes a full data availability statement.

- I could not find a data availability statement in the provided manuscript, is this statement entered in a separate part of the submission process?

- I thank the authors for providing the accompanying code in a [public git repository](<https://github.com/stevenmsm/Joint-COVID-19-and-Influenza-Forecasts-in-the-United-States>), but these materials appear to be incomplete.

I appreciate that it can be difficult and time-consuming to package code and supporting materials so that it is both self-contained and readily usable by others.

My aim in identifying several limitations in the provided repository (listed below) is to support the authors in allowing this study to be reproduced by others.

- The provided `README.md` includes a brief summary of the underlying data, it does not include any user instructions.

- Each `R` source file uses hard-coded directories that are specific to the author's computer (e.g., `~/Documents/Georgia_Tech/Research at GATECH/Research with Dr. Shihao Yang/COVID-19/FLU+COVID19`)

- There are no instructions on how to obtain the necessary data files, how they should be named, and where they should be stored.

Some files are downloaded by the provided scripts (to the hard-coded path shown above) but others are assumed to exist locally.

- Several of the provided source files rely on another file (`ILI_COVID_Data_Clean.R`) which is not included in the repository.

- The provided source files appear to only cover steps 1 and 2 of the method described in section 3.2 of the manuscript, and are missing the third step, which produces the "final winner-takes-all ensemble predictions".

Response to Reviewers Letter for Manuscript “Joint COVID-19 and Influenza Forecasts in the United States using Internet Search Information”

Summary of Major Changes in this Revision:

- We expanded our forecasting and evaluation period by including the most recent data (now from 2020-07-04 to 2022-08-13), according to the suggestion from reviewer #2. All tables and figures in the main draft and supplementary materials are updated accordingly. The overall performance of our national and state-level COVID-19 cases, deaths and %ILI predictions remain consistent and we are able to capture the most recent trend led by Omicron-Variant in early 2022 and summer 2022.
- We incorporate hospitalization as a time series feature for COVID-19 national-level deaths forecasts, according to the suggestion from reviewer #1. All modeling details can be found in the Supplementary Materials. By incorporating hospitalization as the additional time series feature, our COVID-19 deaths predictions remain accurate and robust throughout the changing dynamics with different COVID-19 variants. All modifications are updated in the results section and Supplementary Materials.
- We extend the forecasting horizon for %ILI on both national and state level to 2 weeks, according to the suggestion from reviewer #3. Our 2-week-ahead %ILI predictions remain accurate and robust throughout the evaluation period.
- In addition to the weekly point estimates (forecasts), ARGOX-Joint-Ensemble method now also produces probabilistic forecasts and prediction intervals for all the forecasting targets (COVID-19 cases, deaths, and %ILI), on both national and state-level, according to the suggestion from reviewer #1 and #2. We calculate the 95% nominal prediction interval coverage of the 1-4 weeks ahead COVID-19 cases and deaths and 1-2 weeks ahead %ILI forecasts across 51 regions. We also calculate the weighted interval score (WIS), evaluated across 11 prediction intervals with $\alpha_1=0.02$, $\alpha_2=0.05$, $\alpha_3=0.1$, ..., $\alpha_{11}=0.9$ (implying nominal coverages of 98%, 95%, 90%, ..., 10%), following CDC Forecast Hub's submission guideline [35]. For COVID-19 cases and deaths forecasts, we compare the prediction intervals' coverages and WIS with other publicly available methods submitted to CDC Forecast Hub. For %ILI forecasts, we compare the prediction intervals' coverages and WIS with the lagged 1 vector autoregressive model (VAR). The

prediction intervals' details are included in a newly added section in the Supplementary Materials. All evaluations and analysis are included in the Results section, which still shows competitiveness of our methods in short-term forecasts.

- We simplified the daily ILI data imputation, according to reviewer #1 and #2, by filling the daily ILI with weekly ILI. This change of imputation method still preserves the results and conclusion. All results are updated accordingly. We also preserve the original imputation method as a sensitivity analysis in the Supplementary Materials.
- We modified the layout of our state-by-state COVID-19 cases, deaths and %ILI forecasts evaluations in the Supplementary Material, and added additional details in the Results, and Discussion section to examine the comparing methods' performances during various periods, according to the suggestion of reviewer #2 and #3. We now zoom in on two selected U.S. states, Georgia and North Carolina, for all three forecasting targets, and evaluate the forecasting performances in different covid-19 waves. We notice similar behaviors among the comparing methods in the other states as well.
- We added additional state-level COVID-19 cases and deaths forecasting performance evaluation during three selected rapidly changing dynamics, which correspond to COVID-19 second wave (end of 2020), COVID-19 wave led by Delta variant (summer 2021), and COVID-19 wave led by Omicron variant (early 2022), according to reviewer #2. All methods comparison and analysis are added in Supplementary Material and the Results section.
- We have added additional clarifications in the Introduction and Discussion section according to the suggestion of reviewer #1 and #2.
- We have deposited our organized code (with execution instructions) in Github (<https://github.com/stevenmsm/Joint-COVID-19-and-Influenza-Forecasts-in-the-United-States>), with detailed instructions to execute the code to reproduce the results shown in this study. We have also deposited the online search data used in this study in Harvard dataverse, DOI: 10.7910/DVN/PGNBAX.
- We renamed our proposed method in the revised manuscript in order to clarify and simplify their meanings. Note that we add "single-disease" before a method, indicating that the method only uses one particular disease's information, while adding "bi-disease" before a method, indicating that the method uses both COVID-19 and ILI's information.

For national-level predictions, we renamed the previous “ARGO-Joint” as “ARGO-Nat”, which produces the final national-level forecasts for all targets. For state-level predictions, we renamed the previous “ARGOX-Idv” as “Bi-disease ARGOX-Local”, which is the proposed method in Step 2 (Figure 2). We renamed the previous “Ref [44]” and “Ref [25]” as “Single-disease Ref [44]” and “Single-disease Ref [25]”. All changes are written in color “Blue”. We have also added a section (“Methods Naming Conventions”) in the Supplementary Materials for detailed naming conventions.

Replies to Comments Provided by Reviewer #1

Summary: *The authors aim to build a joint influenza and COVID-19 predictor using search trends for the US to exploit commonalities in symptoms and care seeking for both diseases. Although it's a worthwhile effort, their grasp of existing surveillance streams for these diseases is limited, and there are limited insights derived from similarity in search trends. Further, there are no significant methodological advances other than the choice of exogenous variables. Hence I reject the manuscript in its current form, although I appreciate the authors' effort in exploring this important problem.*

In Figure 1, they primarily compare COVID-19 deaths data with ILI% from a chosen state and its neighbors. First of all, as the authors already acknowledge, the ILI curve could be heavily contaminated by COVID incidence, and hence any observations of similarity could just be due to that. Further, the time period of high similarity seems to be the Omicron wave, where the symptom profile deviated from earlier strains (upper vs lower respiratory, for example - <https://www.nature.com/articles/d41586-022-00007-8>). Also the peak in Fall 2021 coincides with an RSV surge that was seen across multiple regions (<https://www.cdc.gov/surveillance/nrevss/rsv/natl-trend.html>), which is combined with ILI, while the Deaths are due to Delta variant. Without these additional contexts, such correlation based methods could be misled during subsequent variant driven waves of COVID, or off-season resurgence in influenza activity (like being seen in recent weeks Apr-May in USA).

Response: We thank the reviewer for providing detailed and insightful comments. We acknowledge the limitations that the reviewer pointed out. We have revised the paper accordingly (all changes are written in blue), and addressed all concerns and comments raised by

the reviewer via point-to-point responses below. The constructive suggestions have greatly helped us improve the clarity and quality of our draft and are highly appreciated.

We want to emphasize that our proposed model is intentionally a straightforward and principled data-driven approach, leveraging Google search data and inspired by prior influenza research, which prevents us from overfitting. The model is unified for both national and state-level forecasts for both diseases and performs reasonably against baseline models and other publicly available benchmark models. We have added a subsection about “our contribution” in the introduction section, which now reads:

“The ensemble framework is systematic and comprehensive, and each sub-models within the framework is intentionally straightforward and unified to prevent overfitting. Numerical comparisons show that our method performs competitively with other publicly available single-disease forecasting methods. This study further emphasizes the general applicability and the predictive power of online search data for various tasks in disease surveillance.”

As the reviewer mentioned, the similarity between the two diseases is complex [2] and could be heavily contaminated by various factors over time, including other diseases (RSV) and COVID-19 variants. This is one of the biggest challenges in this forecasting task, as we cannot foresee nor manually let the model know all the external factors and possible contamination during the prediction. In addition, we are aware that our estimation targets, JHU COVID-19 dataset (cases, deaths) and CDC’s %ILI, can be unreliable. JHU COVID-19 dataset [42], the curated dataset used by CDC at their official website, retrospectively corrected past confirmed cases and deaths due to reporting error, especially during the early stage of COVID-19. On the other hand, %ILI, released by CDC every week, is the groundtruth for flu and is the percentage of outpatient visits with influenza-like illnesses. This is a proxy for the true flu incidence in the population, as it is calculated from a sample of outpatient visits with influenza-like symptoms. The reported %ILI at state level could have high noise due to its limited sample size and retrospective revisions. This is indeed the limitation of the groundtruth for both COVID-19 and flu, and the surveillance measures. Nevertheless, accurate predictions of COVID-19 cases and death, and CDC’s %ILI at national and state level are valuable for optimizing resource allocations, and healthcare interventions. We have added the following paragraph in the Discussion section, which now reads:

“In addition, it should be noted that our estimation targets (treated as the groundtruth), JHU COVID-19 dataset (cases, deaths) and CDC’s %ILI, can be unreliable. JHU COVID-19 dataset [42] retrospectively corrects past confirmed cases and deaths due to reporting error, especially during the early stage of COVID-19. Furthermore, due to wide availability of rapid antigen tests, the COVID-19 confirmed case count might be an underestimation of the true volume of infections [60]. On the other hand, %ILI is only a proxy for the actual flu incidence in the population, as it is calculated from a sample of outpatient visits with influenza-like symptoms. With symptomatic similarities with COVID-19 and other respiratory pathogens, %ILI could be over-estimating true influenza incidences, especially during the initial outbreak of COVID-19 (early 2020) and the recent COVID-19 Omicron variant outbreak (early 2022 and summer 2022). Furthermore, the reported %ILI at the state-level could have high noise due to its limited sample size and retrospective revisions [25]. Thus, using %ILI as proxy for influenza activities could become less reliable if there were to be new, sudden COVID-19 development. Nevertheless, accurate predictions of COVID-19 cases, deaths, and CDC’s %ILI at both national and state levels are still valuable for optimizing resource allocations, and healthcare interventions, as JHU published COVID-19 case/death counts and CDC published %ILI are still treated as the gold standard by the government and health-care officials. Studies investigating alternative indicators for COVID-19 and flu incidence in the population can be found in [61, 62].”

Comment: *The authors do not seem to exploit the HHS hospitalization dataset (<https://healthdata.gov/Hospital/COVID-19-Reported-Patient-Impact-and-Hospital-Capa/g62h-syeh>) which is the basis for COVID and Influenza hospitalization forecasting exercises being coordinated by CDC. In addition to being directly attributable to respective diseases (unlike ILI), they also suffer from less backfill and are more timely due to their daily update cadence.*

Response: We thank the reviewer for pointing out the lack of hospitalization usage (COVID-19 related daily new hospital admissions). Following the reviewer’s suggestion, we incorporate hospitalization as one of the input features for national-level COVID-19 deaths forecasts, and

slightly modify the input features order for all of our targets. In short, on the national-level, in addition to lagged ILI information, we now use lagged COVID-19 cases information to predict COVID-19 cases, and use lagged COVID-19 deaths and hospitalizations to predict COVID-19 deaths. The input features' modification follows the intuitive timeline of an infected patient's journey from confirmed COVID-19 positive to death. By incorporating hospitalization as the additional time series feature, our COVID-19 deaths predictions remain accurate and robust during rapidly changing dynamics led by COVID-19 variants. The state-level forecasting structures remain the same, as shown in Figure 2. All modifications are updated in the results section and Supplementary Materials.

The updated Section 3.1 National Level now reads:

“That is, we use lagged cases, Google search and ILI information as exogenous variables for COVID-19 case forecasts, while using lagged hospitalization, deaths, Google search and ILI information for COVID-19 death forecasts. Then, we aggregate the daily predictions into future 4 weeks ahead forecasts for reporting and evaluation, consistent with other publicly available benchmark methods.”

Comment: *Further it has been known that recent cases and deaths counts from states for COVID-19 have been heavily influenced by reporting issues. Especially for cases, due to wide availability of rapid antigen tests, it does not truly represent the volume of infections being generated on the ground. Death data has been subject to significant backfill during audits, and are hence unreliable for timely forecasting efforts.*

Response: We thank the reviewer for pointing out the reporting issues in COVID-19 cases and deaths counts. Indeed, the groundtruth for both diseases can be unreliable, and is the inherent limitation of the forecasting tasks. In particular, as the reviewer has mentioned, the COVID-19 cases can be subject to substantial reporting issues with the availability of rapid antigen tests, while COVID-19 deaths might be subject to retrospective revisions. COVID-19 cases is indeed a relative proxy to indicate the infectious status of the general population. Nevertheless, the JHU published COVID-19 confirmed case counts are still used by CDC as the groundtruth [42].

Moreover, by using New York Times github published COVID-19 data (which has no retrospective revisions) as input and focusing on COVID-19 deaths predictions, we are able to

produce accurate forecasts that are valuable for optimizing resource allocations, and healthcare interventions at both national and state level. In the Discussion section, we have added a paragraph regarding the limitations of the gold-standard groundtruth (please also see the Response for the first comment).

We added a sentence in the discussion section that acknowledges this limitation which reads:

“In addition, it should be noted that our estimation targets, JHU COVID-19 dataset (cases, deaths) and CDC’s %ILI, can be unreliable. JHU COVID-19 dataset [42] retrospectively corrects past confirmed cases and deaths due to reporting error, especially during the early stage of COVID-19. Furthermore, due to wide availability of rapid antigen tests, the COVID-19 confirmed case count might be an underestimation of the true volume of infections [60].

Comment: *The authors statement that "if CDC had published daily ILI data, it would have exhibited similar in-week seasonality to COVID-19 reported cases" belies their understanding of the two surveillance systems. While the ILI is obtained via an Outpatient network of hospitals, and includes a numerator (patients with ILI symptoms) and denominator (total outpatient visits), the latter is obtained from State Departments of Health which have their own mechanism for collecting COVID-19 case data from testing centers and hospitals within the state. Further, daily imputation of a % time series like ILI, would involve imputing numerator and denominator separately, to be consistent, which the authors do not do.*

Response: We thank the reviewer for pointing out the potential limitation of our current imputation strategy. Indeed %ILI is the percentage of outpatient visits with influenza-like illnesses, and is computed by dividing the total number of outpatient visits with influenza-like illnesses with the total number of outpatient visits in the region and the timestamp of interest. Our previous imputation mechanism does not impute the numerator and denominator, and is a lack of consideration. Yet, separately imputing numerator and denominator for all states and regions is intractable. Therefore we propose a simpler imputation where the daily ILI number is the same as the weekly ILI number. Essentially, we are now assuming the daily number of patients with ILI symptoms within the week is consistent. This change of imputation method still

preserves the accuracy and robustness of our COVID-19 forecasts for all the targets during the evaluation period from 2020-07-04 to 2022-08-13, shown in the revised results and discussion sections. The imputation method details and results are updated accordingly in the manuscript and Supplementary Materials. The consistency in results regardless of the imputation method also serves as the evidence for the robustness of our study. Part of the previous imputation method and the corresponding result is now moved to the Supplementary Materials section "ILI Imputation Method Sensitivity Analysis", as additional sensitivity analysis for the imputation.

Comment: Finally, as the authors are aware of ForecastHub efforts, providing point estimates for predictions and non-probabilistic error functions (RMSE, MAE, PC) are limited in utility. Especially with the noise in observed data and uncertainty going forward, probabilistic forecasts are more appropriate. I suggest the authors look into <https://journals.plos.org/plosmedicine/article/comments?id=10.1371/journal.pmed.1003793> for future reporting epidemic forecasting tasks. This also makes it difficult to compare against other methods in the Hub, since they are better evaluated using metrics such as weighted interval score (WIS) and coverage.

Response: We thank the reviewer for the suggestion of including probabilistic forecasts. Indeed, probabilistic forecasts can provide additional predictive power and interpretation, especially when the groundtruth can be unreliable. Therefore, in addition to the weekly estimates, the ARGOX-Joint-Ensemble method now also produces prediction intervals for all the forecasting targets (COVID-19 cases, deaths, and %ILI) on both national and state-level.

ARGOX-Joint-Ensemble's prediction interval (PI) is formed by taking the prediction intervals of selected ensemble methods. Tables S12-S14 show the coverage and weighted interval score (WIS) of the prediction intervals across all 51 states for 1-4 weeks ahead COVID-19 cases/deaths predictions and 1-2 weeks ahead %ILI predictions, respectively. The WIS is evaluated across 11 prediction intervals (nominal coverages of 98%, 95%, 90%, ..., 10%), following CDC Forecast Hub's submission guideline [35]. The detailed WIS mathematical formulation can be found in Ref [55]. We additionally compare both prediction intervals' coverages and WIS of COVID-19 cases and deaths with other publicly available methods submitted to CDC Forecast Hub. ARGOX-Joint-Ensemble achieves strong performance for 1

and 2 weeks ahead predictions, but falls short in 3 and 4 weeks ahead predictions as the external signals deteriorate. A portion of the COVID-19 deaths analysis is quoted as follows:

“In addition to the weekly estimates, ARGOX-Joint-Ensemble also gives prediction intervals, as recommended by [56]. ARGOX-Joint-Ensemble’s prediction interval (PI) is formed by taking the prediction intervals of the selected ensemble method. ARGOX-Joint-Ensemble’s prediction interval (PI) corresponds to the selected method from the ensemble. Table S12 shows the prediction interval actual coverage and weighted interval score (WIS) [55], as well as comparisons to other CDC published teams. In summary, our nominal 95% prediction interval has an actual 91% coverage and 89% coverage for 1 and 2 weeks ahead predictions, suggesting reasonable uncertainty quantification albeit slight overconfidence. Our Gaussian-approximated probabilistic forecast is also competitive for 1-2 weeks ahead predictions in WIS compared to other publicly available methods. However, the performance deteriorates as the prediction horizon extends to 3-4 weeks, as we have noted earlier.”

Replies to Comments Provided by Reviewer #2

Summary: *In this analysis, Ma et al. develop and evaluate point-estimate forecasting frameworks that pool information between influenza-like illness (ILI) and COVID-19 surveillance data to produce forecasts for COVID-19 reported cases and deaths and percent ILI from the U.S. ILINet system. The authors retrospectively evaluate performance of these approaches at the national and state-level from early 2020 through early 2022 using RMSE, MAE, and correlation accuracy metrics.*

Response: We thank the reviewer for the summary, and for providing insightful and constructive comments, which greatly assist the draft’s quality and clarity. We have revised the paper accordingly (all changes are written in blue) and addressed all concerns and comments raised by the reviewer via point-to-point responses below.

Comments: *Specific suggestions follow:*

- *From a conceptual point, the authors should be clearer that they are not forecasting influenza but forecasting ILI. ILI is impacted by COVID, influenza, and other respiratory pathogens. Therefore, these are joint forecasts for COVID and ILI, not influenza, and the authors have not supported their statement that there's an "affinity between influenza and COVID-19's growth" but that ILI activity and COVID-19 transmission may be related to each other. In the US since the COVID-19 pandemic began, we have witnessed one season with almost no influenza transmission and one season with historically low influenza transmission. This is likely to have resulted because of behavioral interventions adopted to prevent the spread of COVID-19 and possible viral interference between COVID-19 and influenza.*

Response: We thank the reviewer for pointing out the difference between ILI and influenza. Indeed, the weekly %ILI report published by CDC in different geographical resolutions is the percentage of outpatient visits with influenza-like illnesses, and is a proxy for the true flu incidences in that area. Forecasting ILI is generally (and loosely) referred to as tracking influenza activities in the existing literature [7,12,14,17,22,25]. With symptomatic similarities with COVID-19 and other respiratory pathogens, % ILI could be over-estimating true influenza incidences, especially during the initial outbreak of COVID-19 (early 2020) and the recent COVID-19 Omicron variant outbreak (early 2022 and summer 2022). This is indeed a limitation of the gold-standard and groundtruth. However, government and health-care officials still treat %ILI as the gold standard, and it is still useful to track %ILI for early-detection of influenza outbreaks and to implement corresponding prevention and interventions. In this study, we aim to utilize the affinity between COVID-19 and %ILI to enhance previously proposed single disease forecasting frameworks for joint disease forecasting. We have revised all the wordings accordingly, and all the changes are written in color blue. We have also added a paragraph in the Discussion section regarding the limitations of the gold-standard groundtruth.

"In addition, it should be noted that our estimation targets (treated as the groundtruth), JHU COVID-19 dataset (cases, deaths) and CDC's %ILI, can be unreliable. JHU COVID-19 dataset [42] retrospectively corrects past confirmed cases and deaths due to reporting error, especially during the early stage of COVID-19. Furthermore, due to wide

availability of rapid antigen tests, the COVID-19 confirmed case count might be an underestimation of the true volume of infections [60]. On the other hand, %ILI is only a proxy for the actual flu incidence in the population, as it is calculated from a sample of outpatient visits with influenza-like symptoms. With symptomatic similarities with COVID-19 and other respiratory pathogens, %ILI could be over-estimating true influenza incidences, especially during the initial outbreak of COVID-19 (early 2020) and the recent COVID-19 Omicron variant outbreak (early 2022 and summer 2022). Furthermore, the reported %ILI at the state-level could have high noise due to its limited sample size and retrospective revisions [25]. Thus, using %ILI as proxy for influenza activities could become less reliable if there were to be new, sudden COVID-19 development. Nevertheless, accurate predictions of COVID-19 cases, deaths, and CDC's %ILI at both national and state levels are still valuable for optimizing resource allocations, and healthcare interventions, as JHU published COVID-19 case/death counts and CDC published %ILI are still treated as the gold standard by the government and health-care officials. Studies investigating alternative indicators for COVID-19 and flu incidence in the population can be found in [61, 62].”

- *For these forecasts to be actionable, the authors should provide probabilistic forecasts and measure forecast performance using metrics that assess probabilistic accuracy (like the weighted interval score and prediction interval coverage). Forecasts received by the COVID-19 Forecasting Hub, which the authors compare their forecast to, and CDC's FluSight require probabilistic forecasts.*

Response: We thank the reviewer for the suggestion of including probabilistic forecasts. Indeed, probabilistic forecasts can provide additional predictive power and interpretation, and are required by COVID-19 Forecast Hub. Therefore, in addition to the weekly estimates, the ARGOX-Joint-Ensemble method now also produces prediction intervals for all the forecasting targets (COVID-19 cases, deaths, and %ILI) on both national and state-level. Therefore, in addition to the weekly estimates, the ARGOX-Joint-Ensemble

method now also produces prediction intervals for all the forecasting targets (COVID-19 cases, deaths, and %ILI) on both national and state-level. ARGOX-Joint-Ensemble's prediction interval (PI) is formed by taking the prediction intervals of selected ensemble methods. Tables S12-S14 show the coverage and weighted interval score (WIS) of the prediction intervals across all 51 states for 1-4 weeks ahead COVID-19 cases/deaths predictions and 1-2 weeks ahead %ILI predictions, respectively. The WIS is evaluated across 11 prediction intervals (nominal coverages of 98%, 95%, 90%, ..., 10%), following CDC Forecast Hub's submission guideline [35]. The detailed WIS mathematical formulation can be found in Ref [55]. We additionally compare both prediction intervals' coverages and WIS of COVID-19 cases and deaths with other publicly available methods submitted to CDC Forecast Hub. Our ARGOX-Joint-Ensemble's WIS and coverage have strong performances for 1 and 2 weeks ahead COVID-19 cases and deaths predictions, but fall short in 3 and 4 weeks ahead predictions as the external signals deteriorate. A portion of the COVID-19 deaths analysis is quoted as follows:

“In addition to the weekly estimates, ARGOX-Joint-Ensemble also gives prediction intervals, as recommended by [56].

ARGOX-Joint-Ensemble's prediction interval (PI) is formed by taking the prediction intervals of the selected ensemble method.

ARGOX-Joint-Ensemble's prediction interval (PI) corresponds to the selected method from the ensemble. Table S12 shows the prediction interval actual coverage and weighted interval score (WIS) [55], as well as comparisons to other CDC published teams. In summary, our nominal 95% prediction interval has an actual 91% coverage and 89% coverage for 1 and 2 weeks ahead predictions, suggesting reasonable uncertainty quantification albeit slight overconfidence. Our Gaussian-approximated probabilistic forecast is also competitive for 1-2 weeks ahead predictions in WIS compared to other publicly available methods. However, the performance deteriorates as the prediction horizon extends to 3-4 weeks, as we have noted earlier.”

- *ILI and COVID case and death data can be subject to backfill (i.e. modification of reported data from days to weeks ago). The authors should only utilize the data available at the time the forecasts would be generated to ensure the accuracy reflects “real-time” performance.*

Response: We thank the reviewer for pointing out the potential backfill, and retrospective revision in COVID-19 reportings. Indeed, JHU CSSE COVID-19 dataset [42], a curated dataset used by the CDC at their official website, retrospectively revises itself for past reporting errors or changes in federal and state policies. Therefore, we do not use the JHU COVID-19 dataset as input features. We use the New York Times (NYT) published COVID-19 dataset [40] as input, which does not revise past data, to give more realistic forecasts. We only use the JHU COVID-19 dataset as groundtruth when comparing against other benchmark methods published in CDC COVID-19 Forecast Hub [35]. Both data sources are collected from January 21, 2020 to August 13, 2022. Also, during rolling-window forecast, we only use the input features that are available at the time of forecast. We do not use any forward looking information. We also included a detailed data collection and prediction schedule in the Supplementary Material (section “ARGOX-Joint-Ensemble”) to further clarify our “real-time” prediction generation scheme. Please also see the Response for the next comment for more details on the rolling-window forecasts’ indexing.

Moreover, we have added another paragraph in the discussion section, elaborating on the limitation of the retrospective nature of our study:

“Another limitation is the retrospective nature of this study. Although we are not using any “forward looking” information that wouldn’t be available at the time of prediction to reflect “real-time” performances, the input data sources could still be subject to backfill and revisions. This issue is circumvented for COVID-19 as we use the New York Times (NYT) github COVID-19 dataset [40] as the input to our models, which does not revise past data. Yet, CDC reported %ILI [5] could be subject to revisions, due to reporting delays from health-care providers. However, prior influenza forecasting studies have shown that %ILI

back-fill and modifications typically would not impact forecasting performances too much [58, 59].”

- *It’s unclear from the paper’s methods what forecasting activities could be conducted “in real-time” vs. what could be done retrospectively. For example, would the ILI data used as an input in the COVID case and death forecasting be available at the time forecasts would be generated or does the delay in ILI data publication prevent that? In addition, what data does the “winner takes all” ensemble evaluation need and is that data available in real-time to support that? It’s important that the authors are only using data that would be available at the time the forecasts would have to be made, especially since they are comparing themselves against forecasts in Table 2 that were truly made in “real-time”.*

Response: We thank the reviewer for pointing out the potential confusion of our forecasting activities. We want to emphasize that we are conducting rolling-window forecasts, which means that all the input features we use are those that are available at the time of forecast. We do not use any forward looking information, and restrict our approach similar to those that report to CDC Forecast Hub in “real-time”. We expand this in detail in the Supplementary Materials, Section “Modification of Previously Proposed COVID-19 and Influenza Methods” Subsection “ARGOX-Joint-Ensemble”. A portion of the paragraph now reads:

“Here, we want to state that we are careful on the indexing and are not using any forward looking information, when we use the four sub-models (Figure 2 Step 2) to obtain COVID-19 cases/deaths estimates. As an example, for COVID-19 cases and deaths forecasts, standing on August 13 2022, to produce 1-4 weeks ahead forecasts (August 20 2022, August 27 2022, September 3 2022, September 10 2022), we use COVID-19 cases, deaths, hospitalization, and Google search data up to August 13 2022, and ILI data up to August 5 2022 (since CDC published the %ILI reports for the past week every Friday)... On the other hand, for %ILI, standing on August 12 2022, to produce 1-2 weeks ahead forecasts (August 12 2022 and August 20 2022), we use COVID-19 cases and

Google search data up to August 11 2022, and %ILI data up to August 5 2022 (since CDC published %ILI lag 1 week behind COVID-19 and Google search data, and by the time we predict for August 12 2022, we know those two data source up to August 11 2022).”

- *It's not clear why the authors are imputing daily ILI data when the ILI and COVID forecasting targets being compared are at the weekly scale and how they validated their approach was accurate since reported COVID-19 cases can be impacted by state reporting choices. The authors should justify this approach and ensure the additional complexity added by it is adding something significant to this study.*

Response: We thank the reviewer for pointing out the potential limitation of our current imputation strategy and the lack of clarification. First of all, COVID-19 data are in daily frequency and %ILI are in weekly frequency. Thus, it is intuitive for us to produce daily forecasts for COVID-19 targets and weekly forecasts for %ILI. Also, daily forecasts will enable us with more data, and produce more robust forecasts, especially during the initial outbreak of COVID-19. Yet, CDC Forecast Hub guideline and all the publicly available methods produce weekly forecasts. Thus, for ease of evaluation, we aggregate our daily forecasts into weekly estimations. Essentially, we use overlapping rolling windows, which is now detailly explained in the Supplementary Materials, Section “Daily and Weekly Indexing Details”. A portion of the paragraph now reads:

“Since COVID-19 cases, deaths and related Google search queries are all in daily frequencies, it is intuitive to produce daily forecasts. Thus, for national-level, we use the “ARGO-Nat” method to produce 28-day-ahead daily forecasts, and aggregate them into weekly forecasts to compare with other publicly available methods (following CDC Forecast Hub guidelines [35]). Similarly, on the state-level, in Step 1 (Figure 2), all the raw estimates of COVID-19 cases/deaths are first obtained in daily frequencies and aggregated into weekly estimates. We essentially store them as rolling weekly estimates stored in the daily index. For example, on date August 13, 2022, all raw estimates store the weekly estimate from August 7, 2022 to August 13, 2022; on date August 12, 2022, they store

the weekly estimate from August 6, 2022 to August 12, 2022, etc. By doing this, we are enabled with more data, which is especially helpful for early stage COVID-19 predictions. Thus, all four sub-models also contain weekly estimates but daily indexing.”

Moreover, we simplified our ILI imputation approach. Since %ILI is the percentage of outpatient visits with influenza-like illnesses, and is computed by dividing the total number of outpatient visits with influenza-like illnesses with the total number of outpatient visits in the region and the timestamp of interest, our previous imputation mechanism does not impute the numerator and denominator, and is a lack of consideration. Yet, separately imputing numerator and denominator for all states and regions is intractable. Therefore we propose a simpler imputation where the daily ILI data is the weekly ILI. Essentially, we are now assuming the daily number of patients with ILI symptoms within the week is consistent. This change of imputation method still preserves the accuracy and robustness of our COVID-19 forecasts for all the targets during the evaluation period from 2020-07-04 to 2022-08-13, shown in the revised results and discussion sections. The imputation method details and results are updated accordingly in the manuscript and Supplementary Materials. The consistency in results regardless of the imputation method also serves as the evidence for the robustness of our study. Part of the previous imputation method and the corresponding result is now moved to the Supplementary Materials section “ILI Imputation Method Sensitivity Analysis”, as additional sensitivity analysis for the imputation.

- *Flu activity increased after the end of the study period in the US while COVID activity decreased. The authors should extend the study period to capture this period to see how these methods perform.*

Response: We thank the reviewer for the comment regarding extending the study period. We have expanded our evaluations period from 2020-07-04 to 2022-08-13, by including the most recent data. Therefore, all tables and figures in the main draft and supplementary materials are updated accordingly. The overall performance of our national and state-level 1-4 weeks ahead predictions remain consistent and we are able to capture the most recent trend led by Omicron-Variant in early 2022 and summer 2022.

- *The authors should look at performance of the specific methods during periods of rapidly changing dynamics (e.g. peaking periods, increasing periods) to see how the methods perform when forecasts are most challenging.*

Response: We thank the reviewer for pointing out the emphasis on analyzing specific periods. We have added in additional forecasting performance evaluation and analysis for both COVID-19 cases and deaths during three rapidly changing dynamics. Table S10 shows 1-4 weeks ahead of state-level COVID-19 deaths prediction performance comparisons in three selected rapidly changing dynamics: COVID-19 second wave (2020-11-14 to 2021-03-06), Delta variant (2021-07-31 to 2021-11-06), and Omicron variant (2022-01-01 to 2022-03-19). Table S11 shows 1-4 weeks ahead of state-level COVID-19 cases prediction performance comparisons in three selected rapidly changing dynamics: COVID-19 second wave (2020-10-31 to 2021-02-20), Delta variant (2021-07-17 to 2021-10-23), and Omicron variant (2021-12-18 to 2022-03-05). There is a 2-week lag between the three periods (above) for COVID-19 cases and deaths evaluations, due to the natural lag between COVID-19 cases and deaths waves. For both COVID-19 cases and deaths evaluations, we compare ARGOX-Joint-Ensemble against COVIDhub-ensemble [35], which is the ensemble forecast collected by CDC Forecast Hub, and naive method, which uses current weeks' counts from New York Times (NYT) as next 1-4 weeks estimations. ARGOX-Joint-Ensemble produces strong short-term forecasts and reasonable 3-4 weeks ahead forecasts in all the periods. A portion of COVID-19 deaths' forecasting analysis from the Results section is quoted below.

“Moreover, ARGOX-Joint-Ensemble is able to produce accurate short-term short-term forecasts, and reasonable 3-4 weeks forecasts, during different rapidly changing dynamics when forecasts are most challenging. Table S10 shows the forecasting performances of the comparing methods, averaging over the 51 states for three selected rapidly changing dynamics: COVID-19 second wave (November 2020 to March 2021), COVID-19 Delta variant (July 2021 to November 2021), and COVID-19 Omicron variant (January 2022 to March 2022). By efficiently leveraging the single-disease and bi-disease sub-models,

ARGOX-Joint-Ensemble can steadily outperform the persistence (naive) method, and produce reasonable forecasts compared to COVIDhub-ensemble [35].”

Additionally, we have revised our result analysis accordingly by zooming into two U.S. states, Georgia and North Carolina, and evaluating the rapidly changing dynamics in particular. For example, in the COVID-19 deaths’ forecasting evaluations, we examine ARGOX-Local’s performance against the single-disease model (Ref [44]), and ARGOX-Joint-Ensemble’s pooling (ensemble selection) behavior. A portion of the Results section now reads:

“To further examine the ARGOX-Joint-Ensemble’s performance gain, we zoom in on the forecasting performances of two U.S. states, Georgia (GA) and North Carolina(NC) (Table S16, S17, Figure S12, S13). Notably, the bi-disease ARGOX-Local model for COVID-19 is more accurate during the increasing periods (e.g. COVID-19 death from Jul 2021 to Oct 2021) and peaking periods (e.g. COVID-19 death in early Oct 2021 and early Feb 2022) than the single disease model. On the other hand, the bi-disease ARGOX-Local model could overshoot and have delayed recovery after peaking periods (e.g. late Feb 2022 post Omicron peak), possibly due to the misleading %ILI signal. Luckily, the ensemble framework is able to select the more robust one between the single-disease and the bi-disease sub-models. Specifically in GA, the ARGOX-Joint-Ensemble selects the bi-disease ARGOX-Local during the increasing periods (Jul 2021 to Oct 2021, and Jan 2022 to Mar 2022), while “falling back” to single disease sub-model post-peak periods (Oct 2021 to Dec 2021, and Mar 2022 to May 2022). We observe similar patterns in bi-disease model performances and ensemble selection behaviors in NC (Table S17 Figure S13) as well as other states.”

Replies to Comments Provided by Reviewer #3

Summary: *In this manuscript, the authors claim that (separate) COVID-19 and seasonal influenza forecasts can be improved by incorporating both COVID-19 and influenza data into the forecasting models. To support this claim, the authors introduce several models for forecasting COVID-19 deaths, COVID-19 cases, and CDC influenza-like illness (ILI) data, and evaluate their performance at (USA) state and national levels over the period 2020-07-04 to 2021-03-05. The authors demonstrate that the COVID-19 case forecasts generated by models that incorporate both COVID-19 and influenza data are competitive with the best-performing models in the CDC COVID-19 forecasting ensemble. The simultaneous circulation of COVID-19 and seasonal influenza is a major concern for healthcare systems around the world. We should expect some correlation between COVID-19 cases and influenza-like illnesses, since these pathogens share common modes of transmission, and much of this correlation could presumably be explained by human mobility and mixing. Accordingly, the major claim in this paper is sensible and, as far as I am aware, has not been investigated in the existing literature. The results clearly support the authors' claim, and it is impressive that the proposed "ARGOX-Joint-Ensemble" model is competitive with the best-performing models in the CDC ensemble.*

Response: We thank the reviewer for providing detailed summary and insightful comments. We also thank the reviewer for the encouraging words on this important problem. We have revised the paper accordingly (all changes are written in blue), and addressed all concerns and comments raised by the reviewer via point-to-point responses below. The constructive suggestions have greatly helped us improve the clarity and quality of our draft and are highly appreciated.

Comments:

- 1. The "winner takes all" approach is not described in enough detail to be reproduced. The authors state: "particularly, for a training period of 15 weeks, we evaluate both predictors with mean squared error (MSE) and select the one with lowest MSE as the ensemble predictor for future weeks." While this text makes it clear that the approach selects a single model at each forecast date, and separately for each of the 9 forecast targets (1-4 weeks ahead COVID-19 deaths, 1-4 weeks ahead COVID-19 deaths, 1 week ahead %ILI), the only supporting information I could find is Table S4, which shows how often each competing model was selected for each of the 9 forecast targets.*

At each forecast week, were the previous 15 weeks used as a training period to identify the best model for each forecast target?

If so, how did this account for forecast targets with fewer than 15 data points (e.g., 4 week ahead targets)?

If not, how are the models evaluated?

Response: We thank the reviewer for pointing out the unclear explanation of our “winner takes all” approach. We detailly explain the approach below, and also added a new section (Section “ARGOX-Joint-Ensemble”) in the Supplementary Materials.

At each forecast, we will have four forecast values (one generated from bi-disease ARGOX-Local and three generated from single-disease Ref [44]) for 1-4 weeks ahead COVID-19 cases and deaths. We will have two forecast values (one from bi-disease ARGOX-Local and one from single-disease Ref [25]) for 1-2 week ahead %ILI. For COVID-19 cases/deaths, we determine which of the four forecast values to select as the ARGOX-Joint-Ensemble’s forecast, based on the comparison of mean squared error (MSE) between the four methods on their past 15 weeks’ predictions (of the same target). Note that we flattened out the three sub-models introduced in single-disease Ref [44] in the final ensemble step here to produce a 4-method ensemble forecast as ARGOX-Joint-Ensemble's final forecast, due to simplicity. The results are similar if we conduct a 2-method ensemble forecast (single-disease Ref [44] and bi-disease ARGOX-Local), i.e. not flattening the 3 sub-models. For %ILI, we determine which of the two forecast values to select as the ARGOX-Joint-Ensemble’s forecast, based on the comparison of MSE on their past 15 weeks’ predictions. Thus, the answer to reviewer’s first sub-question is “Yes, we treat the previous 15 weeks forecasts from the four sub-methods separately as training, and compare the four MSE computed in order to select the ensemble forecast for the current target. This is done separately for each of the targets.”

To answer the second question, we have to clarify that the 15 training weeks considered for MSE computation are overlapping weeks if the target is COVID-19 cases/deaths, while non-overlapping if forecasting %ILI.

Therefore, the overlapping 15 weeks training period is essentially a 2 weeks time span. The overlapping 15 weeks’ forecasts seem to be highly correlated. However, with

thorough simulation and experiments, this generates better performances than simply using 2 weeks' (separate) forecast for MSE comparison and ensemble selection.

As an example, “standing on” August 6th 2022 (all input data available up to 8/6/2022), to forecast the 1-week-ahead COVID-19 cases (the week of 8/13/2022), we will have four potential candidates from Ref [43] and bi-disease ARGOX-Local. If bi-disease ARGOX-Local has lower MSE for its previous 15 weeks' 1-week-ahead forecasts, we will select bi-disease ARGOX-Local's forecast value as ARGOX-Joint-Ensemble's forecast value for 8/13/2022 (vise-versa). Here, the 15 weeks' 1-week-ahead forecasts are 8/6/2022, 8/5/2022, ..., 7/23/2022. Since all input data for COVID-19 forecasts are daily, we also forecast in daily frequency and sum into weekly forecasts, i.e. 8/6/2022 stores the weekly estimate from 7/31/2022 to 8/6/2022; 8/5/2022 stores the weekly estimate from 7/30/2022 to 8/5/2022, etc. Therefore, the training period is an overlapping 15 weeks, and we won't encounter cases with less than 15 training data points. On the other hand, since %ILI data dates back to 2014, we won't run into this problem either.

We explain the ensemble approach and the indexing of our methods in the Supplementary Material Section “Modification of Previously Proposed COVID-19 and Influenza Methods” Subsection “ARGOX-Joint-Ensemble”. A portion of the paragraph now reads:

“Lastly, when conducting the ensemble forecast for 1-4 weeks ahead COVID-19 cases/deaths, we use 15 overlapping weeks as training period. For example, “standing on” August 13, 2022 (all sub-model estimates obtained up to August 13, 2022 and all data available up to August 13, 2022), to forecast the 1-week-ahead COVID-19 cases (the week of August 20, 2022), we will have four potential candidates “ARGO”, “ARGOX-2Step”, “ARGOX-NatConstraint” (from single-disease Ref [44]) and “bi-disease ARGOX-Local”. If bi-disease ARGOX-Local has lower MSE for its previous 15 weeks' 1-week-ahead forecasts, we will select bi-disease ARGOX-Local's forecast value as ARGOX-Joint-Ensemble's forecast value for August 13, 2022 (vise-versa). Here, the 15 weeks' 1-week-ahead forecasts are 8/13/2022, 8/12/2022, ... ,

7/30/2022. Therefore, the overlapping 15 weeks training period is roughly a 2-week time span. The overlapping 15 weeks' forecasts seem to be highly correlated, but this generates better performances than using 2 separate weeks' forecasts for MSE comparison and ensemble selection (i.e. keep all raw and sub-model estimates in weekly frequencies)”

2. *Why did the %ILI forecasts only have a 1-week horizon, as compared to the 4-week horizon used for the COVID-19 forecasts?*

I did not notice an explanation for this decision in the provided manuscript (my apologies if the authors have explained their rationale!).

Response: We thank the reviewer for the suggestion of expanding the forecasting horizon for %ILI. We now produce up to 2-week-ahead %ILI forecasts at both national and state level, following previously proposed methods [23, 24, 25] and other data-driven %ILI forecasting studies [12, 26, 29, 39]. Yet, the proposed ARGO-Nat (national-level) and ARGOX-Joint-Ensemble (state-level) gradually lose their predictive power towards %ILI, when forecasting horizon extends to 3 and 4 weeks, due to signal deterioration in influenza related Google search data and COVID-19 confirmed cases, and therefore we focus on 1-2 weeks ahead %ILI predictions in this study.

One hypothesis is that the COVID-19 symptoms and contagious period last longer than influenza [57], and thus the COVID-19 time series and related Google search information are more predictable for COVID-19 than those for %ILI for longer forecasting horizons. We have added the explanation above in the Discussion section. The edited portion of the Discussion section now reads:

“Similarly, the proposed ARGO-Nat (national-level) and ARGOX-Joint-Ensemble (state-level) gradually lose their predictive power towards %ILI, when forecasting horizon extends to 3 and 4 weeks, due to signal deterioration in influenza related Google search data and COVID-19 confirmed cases, and therefore we focus on 1-2 weeks ahead %ILI predictions in this study. One hypothesis is that the COVID-19 symptoms and contagious period last longer than influenza [57], and thus the COVID-19 time series and related Google search information are

more predictable for COVID-19 than those for %ILI for longer forecasting horizons. The long-term forecasts' deterioration is indeed a limitation, due to the data-driven nature of our proposed models, impacting the predictions for the onset and the finish of the disease season”

3. *I'm keen to understand more about how the forecast predictions changed as a consequence of using both COVID-19 and %ILI data, and how this is reflected in the reported performance improvements.*

It's difficult to assess this by eye from the 160 or so figures included in the supplementary materials, but hopefully the authors can address some of the following questions:

- *Are the "ARGOX-Joint-Ensemble" forecasts qualitatively different in some way, as a result of using the pooled COVID-19 and %ILI data?*

Response: We thank the reviewer for pointing out the potential lack of focus in the supplementary material. We have revised the Supplementary Material and focus on two U.S. states, Georgia and North Carolina, for all three forecasting targets. We have also revised the results section accordingly, by evaluating ARGOX-Joint-Ensemble's performance closely against prior single disease models (Ref [44]) and bi-disease ARGOX-Local, in those two states during rapidly changing dynamics.

In short, ARGOX-Joint-Ensemble forecast is qualitatively different and better from previous single disease models, thanks to the pooling of COVID-19 and %ILI information. It is able to robustly select the “best” sub-models among the single disease sub-models (Ref [44]) and bi-disease ARGOX-Local, during increasing, peaking and decreasing periods. It can quickly recognize the overestimation of one and fall back to select the other. In the Result Section, COVID-19 deaths' evaluations, a portion of the paragraphs now reads:

“Luckily, the ensemble framework is able to select the more robust one between the single-disease and the bi-disease sub-models. Specifically in GA, the ARGOX-Joint-Ensemble selects the bi-disease ARGOX-Local during the increasing periods (Jul 2021 to Oct 2021, and Jan 2022 to Mar 2022), while “falling

back” to single disease sub-model post-peak periods (Oct 2021 to Dec 2021, and Mar 2022 to May 2022). We observe similar patterns in bi-disease model performances and ensemble selection behaviors in NC (Table S17 Figure S13) as well as other states.”

→ *Are there certain circumstances or narrow windows in time where the "ARGOX-Joint-Ensemble" forecasts substantially out-perform the non-pooled models, or do they instead consistently perform slightly better than the non-pooled models?*

Response: By zooming into Georgia and North Carolina, we can see that ARGOX-Joint-Ensemble tends to select bi-disease ARGOX-Local more often during increasing and peaking periods (especially for 1-2 weeks ahead forecasts), as bi-disease ARGOX-Local can foresee the upcoming surge earlier than the other single disease sub-models. On the other hand, ARGOX-Joint-Ensemble can also recognize bi-disease ARGOX-Local’s overestimations and fall back to the other sub-models relatively quickly. Thus, ARGOX-Joint-Ensemble forecasts are able to outperform the unpooled models (single disease models) and bi-disease ARGOX-Local on average across all states and all forecasting horizons. However, it cannot uniformly outperform them in all the states and all the time range. Nevertheless, the ensemble framework is robust in its selection and can accurately forecast COVID-19 and ILI jointly.

For example, in the COVID-19 deaths evaluation for North Carolina, the section “Detailed COVID-19 death estimation results for each state” in the Supplementary Material now reads:

“In North Carolina, ARGOX-Joint-Ensemble's selection is similar, and it is also able to recognize the “best” sub-model. For example, during the increasing period from January 2022 to March 2022, for 2 weeks ahead forecasts, ARGOX-Local estimates the increasing trend too aggressively and therefore ARGOX-Joint-Ensemble selects the sub-models from single-disease Ref [44] instead. As ARGOX-Local's performances

gradually deteriorate 3 and 4 weeks ahead of forecasts, ARGOX-Joint-Ensemble can quickly recover and select the alternative sub-models instead (Table S4 and Figure S12, S15)."

→ *Under what circumstances did the "ARGOX-Joint-Ensemble" consistently select the pooled models, and under what circumstances did it consistently select the non-pooled models?*

Response: Similar to the response above, bi-disease ARGOX-Local can foresee the upcoming surge earlier than the other single disease sub-models, especially for 1-2 weeks ahead forecasts, due to its ability to capture neighboring states' ILI and COVID-19 growth trends. However, ILI signal has relatively short term time span, and could negatively impact bi-disease ARGOX-Local 3-4 weeks ahead forecasts. Thus, ARGOX-Joint-Ensemble tends to select bi-disease ARGOX-Local (pooled model) more often during increasing and peaking periods, and non-pooled models during decreasing periods and 3-4 weeks ahead COVID-19 forecasts.

→ **How rapidly* can the "ARGOX-Joint-Ensemble" forecasts recognise when to switch between the pooled and unpooled models?
If the authors used a 15-week evaluation period to select the best model for each forecast target, did they also consider shorter evaluation periods to allow the ensemble to switch more rapidly between the competing models?*

Response: We thank the reviewer for pointing out the possible result difference using different "hyperparameter" values. We apologize again for the unclarity regarding the training period of "winner-takes-all" step. Since we are using 15 overlapping weeks as training to select the ensemble forecast for COVID-19, this essentially means the training period is roughly 2 weeks, which is able to capture sudden changes in the target and adjust the forecast accordingly. With thorough simulation and experiments, the overlapping 15 weeks for COVID-19 forecasts generate better performances than simply using 2 weeks' (separate) forecast for MSE comparison and ensemble selection, as well as other ensemble training

lengths. The non-overlapping 15 weeks works the best for %ILI. The same hyperparameter choices for both diseases' forecasts also unifies the story.

We included more explanations on the ensemble approach in the Supplementary Material (Section "ARGOX-Joint-Ensemble").

4. *In essence, the questions in the previous comment are all aimed at understanding when, and to what degree, we should place confidence in these forecasts.*

An even tougher question to answer, but for which I'm curious to hear the authors' thoughts, is whether the improvements obtained from the "ARGOX-Joint-Ensemble" model can offer earlier warning (or some other form of enhanced information) for healthcare providers?

In particular, for infectious diseases, it can be particularly useful to predict when activity will peak and begin to decrease (and, alternatively, when activity will bottom-out and begin to increase). Have the authors assessed whether the "ARGOX-Joint-Ensemble" model provides better and/or earlier predictions of when there will be a turning point in the data?

Response: We thank the reviewer for pointing out the potential early-warning "feature". Indeed, one of the eventual goals of forecasting tasks is to give early-warnings for health care intervention, and resource allocation. Indeed, with the help of bi-disease ARGOX-Local (pooled-model), ARGOX-Joint-Ensemble is robust towards detecting upcoming surges, especially for 1-2 weeks ahead short term forecast, and is less prone to overfitting and overestimation during peaking periods. However, due to Google search and ILI's signal deterioration, ARGOX-Joint-Ensemble gradually loses its predictive power towards COVID-19 when the forecast horizon extends (similar to %ILI). Nevertheless, ARGOX-Joint-Ensemble can still uniformly outperform baseline time series benchmark and persistence model on average, and perform reasonably against other CDC published COVID-19 forecasts. Furthermore, ARGOX-Joint-Ensemble now also produces prediction intervals (Tables S12-S14), and has robust coverage for 1-2 weeks ahead forecasts, compensating its long-term forecasts' deterioration.

The long-term forecasts' deterioration is indeed a limitation, shown in both point estimates (Tables 1-5) and prediction intervals (Table S12-S14). We cannot predict the

onset and the finish of the disease season, due to the data-driven nature of the models, and therefore no significant long-term insights (such as peak timing, onset of a wave, etc.). We discussed this limitation in our discussion, which reads:

“Like all big-data-based models, our model has its limitations. ARGO-Nat and ARGOX-Joint-Ensemble’s accuracy depends on the reliability and stability of its inputs – Google Trends data, historical %ILI data from CDC, and COVID-19 cases/deaths data from NYT. Since the optimal lags between COVID-19 cases/death and Google search queries have short time-span (Table S1), information in Google search data deteriorates as forecast horizons expand, which could potentially impact the robustness and accuracy of our 3 and 4 weeks ahead COVID-19 predictions. Similarly, the proposed ARGO-Nat (national-level) and ARGOX-Joint-Ensemble (state-level) gradually lose their predictive power towards %ILI, when forecasting horizon extends to 3 and 4 weeks, due to signal deterioration in influenza related Google search data and COVID-19 confirmed cases, and therefore we focus on 1-2 weeks ahead %ILI predictions in this study...”

Comments (minor):

1. *It would be useful to highlight the study period in Figure 1. The authors might also consider including similar plots for the national data?*

Response: We have added the study period of Figure 1 (from 2020-07-04 to 2022-08-13). Due to space limitations, we will omit the similar plot as Figure 1 for national data in this paper.

2. *It would be interesting to include the "ARGOX Local" results in Tables 1-4, so that the reader can see a more nuanced comparison of the incremental gains provided by the pooled data ("ARGOX Local") and the "winner takes all" approach ("ARGOX-Joint-Ensemble"), without resorting to the supplementary material.*

Response: We have now included bi-disease ARGOX-Local in the comparison tables and figures. We also organized the comparing methods (in Tables 1, 3, 5) in bullet-points, according to the ensemble's hierarchy (see Figure 2).

3. *A table of contents would help the reader to navigate the 172-page supporting information document. It appears the authors intended to include a table of contents, since the first page contains the statement "This Supplementary Material is organized as following:", but this is not followed by a table or list.*

Response: We thank the reviewer for the suggestion, and apologize again on the lack of organization in the Supplementary Materials on state-by-state evaluations. We have added a table of contents in the first page of the Supplementary Materials. We also now zoom into two U.S. states, Georgia and North Carolina, for all three forecasting targets, and evaluate their performances and ensemble pooling behavior in different rapidly changing dynamics. Note that the comparing methods have similar behaviors in the other states. All changes are updated in the Results and Supplementary Materials accordingly.

4. *Am I correct in understanding that the lags in Table S3 are reported in days*

Response: Yes, Table S3 demonstrates the optimal lag in days between each important Google search queries and COVID-19 cases and deaths trends.

Comments: *Reproducibility and availability of materials.*

In the Editorial Policy Checklist, the authors have indicated that all relevant code is provided and that the manuscript includes a full data availability statement.

- *I could not find a data availability statement in the provided manuscript, is this statement entered in a separate part of the submission process?*

Response: We apologize for not having the data availability statement. We have added data availability and code availability statements after Discussion section and before references.

- *I thank the authors for providing the accompanying code in a [public git repository](<https://github.com/stevenmsm/Joint-COVID-19-and-Influenza-Forecasts-in-the-United-States>), but these materials appear to be incomplete.*

Response: We have uploaded all the code needed as well as data needed, to reproduce the results shown in this study.

- *I appreciate that it can be difficult and time-consuming to package code and supporting materials so that it is both self-contained and readily usable by others. My aim in identifying several limitations in the provided repository (listed below) is to support the authors in allowing this study to be reproduced by others.*

Response: We thank the reviewer for the suggestion and we apologize for the poor organization of the code and data. We have now organized the code and data accordingly on github and Harvard Dataverse.

- *The provided `README.md` includes a brief summary of the underlying data, it does not include any user instructions.*

Response: We have added the user instructions accordingly in the github README.md.

- *Each `R` source file uses hard-coded directories that are specific to the author's computer (e.g., `~/Documents/Georgia_Tech/Research at GATECH/Research with Dr. Shihao Yang/COVID-19/FLU+COVID19`)*

Response: We have now organized the code accordingly, and removed all the hard-coded directories.

- *There are no instructions on how to obtain the necessary data files, how they should be named, and where they should be stored. Some files are downloaded by the provided scripts (to the hard-coded path shown above) but others are assumed to exist locally.*

Response: We have added the instructions in the README.md. For example, “true-incident hospitalization.csv” stores the daily new hospital admissions in all U.S. states, and is obtained from

“<https://github.com/reichlab/covid19-forecast-hub/tree/master/data-truth>” (the official

CDC Forecast Hub Github Repo use as their groundtruth). “Population.csv” stores the U.S. state census information and is obtained from “<https://www.census.gov>”.

- *Several of the provided source files rely on another file (‘ILI_COVID_Data_Clean.R’) which is not included in the repository. The provided source files appear to only cover steps 1 and 2 of the method described in section 3.2 of the manuscript, and are missing the third step, which produces the “final winner-takes-all ensemble predictions”.*

Response: We have uploaded ILI_COVID_Data_Clean.R, which essentially downloaded ILI data from CDC and COVID-19 data from New York Times, and organized them accordingly. We also organized the code to combine all steps into a single R file for each target. For COVID-19 cases, for example, all three steps are in COVID_Case.R, with detailed comments along with the code.

Reviewers' comments:

Reviewer #2 (Remarks to the Author):

The authors have addressed most of the concerns raised in the previous review, and I greatly appreciate the amount of effort that went into the revision. However, I do think it's critical that the authors reframe this analysis as joint forecasts for COVID-19 and ILI (not influenza). While ILI was one of the main metrics for tracking influenza activity prior to the COVID-19 pandemic, that is no longer the case. Even the CDC website now includes this disclaimer for the ILI data:

"The U.S. Outpatient Influenza-like Illness Surveillance Network (ILINet) monitors outpatient visits for respiratory illness referred to as influenza-like illness [ILI (fever plus cough or sore throat)], not laboratory-confirmed influenza, and will therefore capture respiratory illness visits due to infection with any pathogen that can present with similar symptoms, including influenza, SARS-CoV-2, and RSV. Due to the COVID-19 pandemic, health care-seeking behaviors have changed, and people may be accessing the health care system in alternative settings not captured as a part of ILINet or at a different point in their illness than they might have before the pandemic. Therefore, it is important to evaluate syndromic surveillance data, including that from ILINet, in the context of other sources of surveillance data to obtain a complete and accurate picture of influenza, SARS-CoV-2, and other respiratory virus activity."

Because of the above, the CDC-led forecasting activities switched their forecast target from ILI to flu-associated hospitalizations. Therefore, the authors need to be clear that their forecast target is not an indicator just for influenza that COVID-19 can contribute significantly to the signal. This is best illustrated by the last season in the US when ILI was much higher in Winter 2021 than in Spring 2022, even though there was more flu activity in spring than winter.

Reviewer #3 (Remarks to the Author):

The authors have thoroughly addressed all of my major comments, both in their response letter and in the revised manuscript and accompanying materials. I thank them for their detailed responses, which have addressed my concerns and reservations.

1. The third and final methods step is now clearly explained in the supplementary materials, and flagged in section 3.2 of the manuscript.
2. The authors have added a 2-week %ILI forecast horizon, and highlight that the predictive power decreases for longer (3-week and 4-week) horizons in the discussion section.
3. The authors have evaluated the ensemble forecasts against the single-disease and bi-disease forecasts, with a focus on time periods with distinct epidemic characteristics, in sections 4.1, 4.2, and 4.3. They have also substantially revised the supplementary materials, which are greatly improved in clarity and focus.
4. The authors have addressed the limitations related to longer-term insights in the discussion section.

The authors have provided detailed, thorough responses to each of my minor comments. They have also greatly improved the accompanying code and documentation in the public git repository. I thank them for their careful consideration and attention to detail.

I have two minor comments regarding the revised manuscript and materials:

1. With the (welcome) addition of prediction intervals, it might be worth including an example plot of the ensemble prediction intervals (e.g., COVID-19 deaths for Georgia and/or North Carolina) in the supplementary materials, if not in the main manuscript itself.

2. Some of the file paths in the accompanying code are specified relative to the user's home directory (`~`) and would be more appropriately defined relative to the working directory (`. `). For example, the path for `Populations.csv` is defined as `~/Population.csv` in `COVID19_Case.R`, `COVID19_Death.R`, and `ILI_COVID_Data_Clean.R`.

Response to Reviewers Letter for Manuscript “Joint COVID-19 and Influenza Forecasts in the United States using Internet Search Information” - Round 2

Summary of Major Changes in this Revision:

- We have now reframed all analysis from Influenza forecasts to Influenza-like Illness forecasts, and added additional limitations and future work descriptions in the Discussion section, according to reviewer 2 and 3’s comments.
- We have included visualizations for ARGOX-Joint-Ensembles’ probabilistic predictions of COVID-19 deaths in Georgia and North Carolina, according to the suggestion from reviewer 3.
- We have included detailed descriptions of ARGOX-Joint-Ensemble’s probabilistic predictions in the Supplementary Materials, according to the suggestion from the reviewer 3.

Replies to Reviewer 3 based on our previous rebuttal to Reviewer 1 (per Editor’s request):

Comment: In addition to commenting on your rebuttal to their comments, Reviewer 3 also responded confidentially to a request from me to comment on your rebuttal to Reviewer 1. They commented that most requests have been adequately addressed however they consider "%ILI could be over-estimating true influenza incidence" to be misleading, because while %ILI is not specific to influenza, clinical presentations are a relatively small proportion of influenza infections. Also, the trend in %ILI may not be indicative of influenza activity, due to the circulation of other pathogens that cause ILI. This should be considered in the discussion about the implications of the results, but obviously does not affect how you measure and interpret the ability of your model to forecast the data.

Response: We thank the reviewer for the insightful comment on the forecasting target descriptions of the current manuscript. Indeed, in light of the reviewer 2’s comment below and the current CDC FluView’s descriptions of outpatient respiratory illness surveillance, we have now reframed the manuscript’s analysis from Influenza forecasts to Influenza-like Illness forecasts, which includes the following changes:

- We have changed all wordings, subject to the CDC FluView’s descriptions on ILI. For instance, we changed all “Influenza” to “Influenza-like Illness”.
- We have rephrased the CDC FluView’s descriptions and added additional limitation descriptions in the Discussion section. A portion of the revised paragraph is quoted below:

“On the other hand, %ILI is only a proxy for the actual flu incidence in the population. First of all, %ILI could exhibit high noise at the state-level, as it is calculated from a sample of outpatient visits with influenza-like symptoms and subjects to retrospective revisions [24]. Moreover, with the prolonged COVID-19 pandemic and subsequent changes in public’s health care-seeking behaviors, %ILI is no longer accurately indicative of laboratory confirmed influenza, as it will capture visits due to any respiratory pathogen that presents with the symptoms of fever plus cough or sore throat, including influenza, SARS-CoV-2 (COVID-19), and Respiratory Syncytial Virus (RSV) [60]. Nevertheless, under the current pandemic situation, accurate predictions of COVID-19 cases, deaths, and %ILI at both national and state levels are still valuable for optimizing resource allocations, and healthcare interventions. For example, the %ILI surveillance data can still reveal the general trend of influenza activity in a particular region and provide invaluable information for optimizing where, when and what influenza viruses are circulating [60].”

- With the changes from the above bullet-point. The sentence in the previous manuscript version, “%ILI could be over-estimating true influenza incidence.”, is removed.
- We have added additional descriptions for future directions in the Discussion section, addressing CDC FluSight’s current forecasting target. The edited portion of the Discussion section is quoted below:

“In addition, CDC FluSight is also investigating additional surveillance components to track seasonal influenza activities, including laboratory-confirmed influenza hospital admissions [63]. Therefore, considering alternative influenza activities’ indicators as forecasting

targets and/or exogenous information in the model could be an important future direction.”

Comment: *Also, this reviewer commented that it was not clearly stated how you calculated the prediction intervals. This should be added.*

Response: We thank the reviewer for the suggestion of including more details in the prediction interval calculation. We have now added more detailed descriptions of ARGOX-Joint-Ensemble’s prediction interval calculation and ARGOX-Local’s prediction interval calculation in the Supplementary Materials. In particular, the added description of ARGOX-Local’s prediction interval calculation is quoted below:

“To obtain the prediction interval of $\hat{y}_{\tau,m}$, we first estimate the variance of the prediction residual empirically. In particular, we estimate $Var(y_{\tau,m} - \hat{y}_{\tau,m} | y_{\tau-1,m}, y_{\tau-2,m}, \dots)$ using the most recent 2-month training window’s empirical residuals. Then, the corresponding $1 - \alpha\%$ prediction interval estimate is as follows: $\hat{y}_{\tau,m} \pm z_{1-\alpha} \sqrt{Var(y_{\tau,m} - \hat{y}_{\tau,m} | y_{\tau-1,m}, y_{\tau-2,m}, \dots)}$ ”

The added description of ARGOX-Joint-Ensemble’s prediction interval calculation is quoted below:

“Lastly, ARGOX-Joint-Ensemble’s prediction interval for state m at week t directly corresponds to the selected method’s prediction interval in the state and week of interest. The detailed descriptions of the “ARGO”, “ARGOX-2Step”, and “ARGOX-Nat-Constraint” prediction interval calculations can be found in Ref [43]. On the other hand, “ARGOX-Local” prediction interval is computed empirically, and is described in the section “Newly Proposed Bi-disease ARGOX-Local” above. As an example, if ARGOX-Joint-Ensemble selected “ARGOX-Local” as the “best” predictor for week 8/20/2022 in Georgia, the prediction interval will be calculated by “ARGOX-Local”.”

Comment: *They also noticed that references 22 and 23 are identical.*

Response: We thank the reviewer for pointing out the duplicated references. We have now removed one of them.

Replies to Comments Provided by Reviewer #2:

Summary: *The authors have addressed most of the concerns raised in the previous review, and I greatly appreciate the amount of effort that went into the revision.*

Response: We thank the reviewer for providing detailed and insightful comments. We appreciate the thorough and insightful comments from the reviewer, which greatly helped us to improve the clarity and quality of the manuscript. We have revised the paper accordingly (all changes are written in blue), and addressed all concerns and comments raised by the reviewer via point-to-point responses below.

Comment: *However, I do think it's critical that the authors reframe this analysis as joint forecasts for COVID-19 and ILI (not influenza). While ILI was one of the main metrics for tracking influenza activity prior to the COVID-19 pandemic, that is no longer the case. Even the CDC website now includes this disclaimer for the ILI data:*

“The U.S. Outpatient Influenza-like Illness Surveillance Network (ILINet) monitors outpatient visits for respiratory illness referred to as influenza-like illness [ILI (fever plus cough or sore throat)], not laboratory-confirmed influenza, and will therefore capture respiratory illness visits due to infection with any pathogen that can present with similar symptoms, including influenza, SARS-CoV-2, and RSV. Due to the COVID-19 pandemic, health care-seeking behaviors have changed, and people may be accessing the health care system in alternative settings not captured as a part of ILINet or at a different point in their illness than they might have before the pandemic. Therefore, it is important to evaluate syndromic surveillance data, including that from ILINet, in the context of other sources of surveillance data to obtain a complete and accurate picture of influenza, SARS-CoV-2, and other respiratory virus activity.”

Response: We thank the reviewer for the suggestion of reframing the analysis from influenza forecast to ILI forecast, and citing CDC FluView’s descriptions on outpatient respiratory illness

surveillance. We have now reframed the manuscript's analysis to Influenza-like Illness forecasts, which includes the following changes:

- We have changed all wordings, subject to the CDC FluView's description above. For instance, we changed all "Influenza" to "Influenza-like Illness".
- We have rephrased the CDC FluView's descriptions and added additional limitation descriptions in the Discussion section. A portion of the revised paragraph is quoted below:

"On the other hand, %ILI is only a proxy for the actual flu incidence in the population. First of all, %ILI could exhibit high noise at the state-level, as it is calculated from a sample of outpatient visits with influenza-like symptoms and subjects to retrospective revisions [24]. Moreover, with the prolonged COVID-19 pandemic and subsequent changes in public's health care-seeking behaviors, %ILI is no longer accurately indicative of laboratory confirmed influenza, as it will capture visits due to any respiratory pathogen that presents with the symptoms of fever plus cough or sore throat, including influenza, SARS-CoV-2 (COVID-19), and Respiratory Syncytial Virus (RSV) [60]. Nevertheless, under the current pandemic situation, accurate predictions of COVID-19 cases, deaths, and %ILI at both national and state levels are still valuable for optimizing resource allocations, and healthcare interventions. For example, the %ILI surveillance data can still reveal the general trend of influenza activity in a particular region and provide invaluable information for optimizing where, when and what influenza viruses are circulating [60]."

Comment: *Because of the above, the CDC-led forecasting activities switched their forecast target from ILI to flu-associated hospitalizations. Therefore, the authors need to be clear that their forecast target is not an indicator just for influenza that COVID-19 can contribute significantly to the signal. This is best illustrated by the last season in the US when ILI was much higher in Winter 2021 than in Spring 2022, even though there was more flu activity in spring than winter.*

Response: We thank the reviewer for pointing out that CDC FluSight is now switching the forecasting target to new weekly laboratory-confirmed influenza hospital admissions. We have now re-worded all descriptions and analysis of our forecasting target: ILI (see above comment's response). Additionally, we have added additional descriptions for future directions in the Discussion section, addressing CDC's switch in forecasting target. The edited portion of the Discussion section is quoted below:

“In addition, CDC FluSight is also investigating additional surveillance components to track seasonal influenza activities, including laboratory-confirmed influenza hospital admissions [63]. Therefore, considering alternative influenza activities' indicators as forecasting targets and/or exogenous information in the model could be an important future direction.”

Replies to Comments Provided by Reviewer #3:

Summary: *The authors have thoroughly addressed all of my major comments, both in their response letter and in the revised manuscript and accompanying materials. I thank them for their detailed responses, which have addressed my concerns and reservations.*

- 1. The third and final methods step is now clearly explained in the supplementary materials, and flagged in section 3.2 of the manuscript.*
- 2. The authors have added a 2-week %ILI forecast horizon, and highlight that the predictive power decreases for longer (3-week and 4-week) horizons in the discussion section.*
- 3. The authors have evaluated the ensemble forecasts against the single-disease and bi-disease forecasts, with a focus on time periods with distinct epidemic characteristics, in sections 4.1, 4.2, and 4.3. They have also substantially revised the supplementary materials, which are greatly improved in clarity and focus.*
- 4. The authors have addressed the limitations related to longer-term insights in the discussion section.*

The authors have provided detailed, thorough responses to each of my minor comments. They have also greatly improved the accompanying code and documentation in the public git repository. I thank them for their careful consideration and attention to detail.

Response: We thank the reviewer for providing detailed summary and insightful comments. We also thank the reviewer for the encouraging words on our previous major revision. The constructive suggestions and the minor comments have greatly helped us to improve the clarity and quality of our draft. We have further revised the paper accordingly (all changes are written in blue), and addressed all the minor comments raised by the reviewer via point-to-point responses below.

Comment: *I have two minor comments regarding the revised manuscript and materials:*

1. *With the (welcome) addition of prediction intervals, it might be worth including an example plot of the ensemble prediction intervals (e.g., COVID-19 deaths for Georgia and/or North Carolina) in the supplementary materials, if not in the main manuscript itself.*

Response: We thank the reviewer for the suggestion of including the prediction interval plots. We have now included the visual illustrations of ARGOX-Joint-Ensemble’s point estimates and prediction intervals for COVID-19 death trends in Georgia and North Carolina in the Supplementary Materials. Specifically, Table S17 and S19 show the 1-4 weeks ahead Georgia and North Carolina’s COVID-19 deaths prediction intervals’ performance comparisons between ARGOX-Joint-Ensemble and COVIDhub-ensemble [34] (due to space limitations), through the two probabilistic error metrics: weighted interval score (WIS) and empirical coverage. Figure S13 and S15 visualize the ARGOX-Joint-Ensemble’s point estimates (in color “Blue”) against the groundtruth collected from JHU CSSE COVID-19 dataset [41] (in color “Black”), as well as ARGOX-Joint-Ensemble’s prediction intervals (filled with color “Orange”), for 1-4 weeks ahead Georgia and North Carolina’s COVID-19 death trends. We also added a paragraph to briefly analyze the visualizations, in the Supplementary Materials section “Detailed COVID-19 death estimation results for Georgia and North Carolina”. The Results section is also revised accordingly, which is also quoted below:

“Table S17 and S19 further show the prediction intervals’ empirical coverage and WIS comparisons to COVIDhub-ensemble [34], zooming into Georgia and North Carolina. Figure S13 and S15 visualize ARGOX-Joint-Ensemble’s death prediction intervals in the two states,

respectively. ... Zooming into Georgia and North Carolina, we demonstrate the robustness of ARGOX-Joint-Ensemble's interval predictions in face of rapidly changing disease dynamics."

- 2. Some of the file paths in the accompanying code are specified relative to the user's home directory (~) and would be more appropriately defined relative to the working directory (.). For example, the path for `Populations.csv` is defined as `~/Population.csv` in `COVID19_Case.R`, `COVID19_Death.R`, and `ILI_COVID_Data_Clean.R`.*

Response: We thank the reviewer for the detailed suggestion on the clarity of the code, and we apologize for not being thorough and careful when we published our code. We have now changed all hard-coded directory/paths to "." instead of "~", in all the files in the Github repository.

REVIEWERS' COMMENTS:

Reviewer #2 (Remarks to the Author):

The authors have sufficiently addressed my concerns. Thank you!

Reviewer #3 (Remarks to the Author):

I thank the authors for their detailed responses to my previous comments, and for documenting how the prediction intervals were calculated. In response to this revised manuscript, I have a few minor comments

1. Regarding the reframing of this analysis as joint forecasts for COVID-19 and ILI, I suggest that the authors should also replace the red "Influenza" title in Figure 2 with "ILI".
2. On a related note, on page 13 the authors write "%ILI is no longer accurately indicative of laboratory confirmed influenza ...". However, this limitation of %ILI predates the COVID-19 pandemic; it has always captured visits due to pathogens such as RSV.
3. Are the estimated variances used to define the prediction intervals (equation S9) calculated separately for lead time? I would expect the 4-week-ahead prediction intervals to be substantially wider than the 1-week-ahead prediction intervals, but it isn't clear to me from looking at figures S13 and S15 whether this is the case. Given the exponential nature of the data, I would recommend in future studies that the authors log-transform the data before calculating the prediction intervals.

Response to Reviewers Letter for Manuscript “Joint COVID-19 and Influenza Forecasts in the United States using Internet Search Information” - Round 3

Replies to Reviewer 3:

Summary: *I thank the authors for their detailed responses to my previous comments, and for documenting how the prediction intervals were calculated. In response to this revised manuscript, I have a few minor comments*

Response: We thank the reviewer for the insightful comments. We have revised the manuscript and addressed the first two points, per editor’s suggestion.

Comment: *Regarding the reframing of this analysis as joint forecasts for COVID-19 and ILI, I suggest that the authors should also replace the red "Influenza" title in Figure 2 with "ILI".*

Response: We thank the reviewer for pointing out the mistakes in our Figure 2. We have now changed the “Influenza” to “ILI” in the flowchart (Figure 2).

Comment: *On a related note, on page 13 the authors write "%ILI is no longer accurately indicative of laboratory confirmed influenza ...". However, this limitation of %ILI predates the COVID-19 pandemic; it has always captured visits due to pathogens such as RSV.*

Response: We thank the reviewer for the suggestion in the wording of %ILI not being accurate indicator of laboratory confirmed influenza. We have slightly revised this sentence and quoted below.

“Moreover, the prolonged COVID-19 pandemic and subsequent changes in public's health care-seeking behaviors further impact %ILI's representation power of laboratory confirmed influenza, as it will capture visits due to any respiratory pathogen that presents with the symptoms of fever plus cough or sore throat, including influenza, SARS-CoV-2 (COVID-19), and Respiratory Syncytial Virus (RSV) [60].”